# Mechanistic insights into PCBP1-driven unfolding of selected i-motif DNA at G₁/S checkpoint

Pallabi Sengupta [1], Natacha Gillet [2], Ikenna Obi [1] & Nasim Sabouri [1,3] ✉

I-motifs are non-canonical, four-stranded DNA structures in cytosine-rich genomic regions, yet their protein-mediated regulation remains underexplored. Here, we identify PCBP1 (Poly(rC)-binding protein 1) as a selective i-motif-binding protein that unfolds specific i-motifs depending on their protonation and hairpin-forming propensities. Systematic truncation reveals that individual K-homology (KH) domains of PCBP1 cannot selectively bind or unfold i-motifs, but their coordinated actions restore wild-type PCBP1 functions. Using biochemical, biophysical, and molecular dynamics studies, we demonstrate that KH1+2 domains remodel i-motifs, recruiting KH3 to facilitate unfolding and efficient DNA replication. Chromatin and cell-based investigations reveal that PCBP1-knockdown increases i-motif formation at specific genomic loci, coinciding with G₁/S arrest and elevated γH2AX, indicative of genomic instability. During G₁/S transition, PCBP1 occupancy peaks at these i-motif loci, ensuring i-motif resolution in early S phase. These findings establish PCBP1 as a critical regulator of i-motif dynamics, directly linking its unfolding activity to G₁/S transition and genome stability.

Protein-DNA interactions regulate essential processes like transcription, replication, and DNA repair. These interactions often induce conformational changes in DNA[1–3], leading to the emergence or resolution of various DNA structures, like four-stranded G-quadruplexes (G4s) or intercalated-motifs (i-motifs). Their spatiotemporal dynamics and processing by specialized proteins maintain genomic integrity across diverse organisms[4,5].

I-motifs are unique non-canonical DNA structures found within cytosine (C)-rich sequences, in the complementary strands of guanine (G)-rich G4-forming regions[6,7]. They contain two parallel-stranded duplexes intercalating in an anti-parallel manner via hemi-protonated C:CH⁺ base pairs (Fig. 1A), predominantly under acidic conditions[8,9]. Their biological relevance is controversial due to their requirement of non-physiological acidic pH. However, research has advanced the cellular study of i-motifs by developing iMab antibody[10,11], which specifically detects i-motifs in cellular nuclei. Further, growing evidence of i-motif formation at physiological pH[12–14] within live cells and their

distinct distribution in cancer versus normal cells[15,16] renewed interest in their biological roles.

A significant proportion of i-motif-forming sequences (iMfs) cluster within oncogene promoters[15], where they may modulate transcription by influencing G4 folding on the complementary strands[17]. Moreover, i-motifs form transiently during DNA replication, posing topological obstacles that, if unresolved, may cause deletion and genomic instability[16–18]. In a study by Richter's group using Cut-and-Tag sequencing[15], i-motifs are reported to play significant roles in cell cycle regulation. Similarly, Christ's group evidenced i-motif's peak abundance in the nuclei during G₁/S transition, followed by their resolution during replication initiation in early S phase[10]. This temporal pattern suggests a correlation between i-motif dynamics and cell cycle, implicating specialized proteins capable of stabilizing and/or resolving i-motifs to ensure the precision and fidelity of cellular activities.

Despite the growing interest in i-motif research, the identification of proteins that bind and regulate i-motifs remains limited due to the

[1]Department of Medical Biochemistry and Biophysics, Umeå University, Umeå, Sweden. [2]CNRS, ENS de Lyon, Laboratoire de Chimie, UMR 5182, 46 Allée d'Italie, F-69342 Lyon, France. [3]Science for Life Laboratory, Umeå University, Umeå, Sweden. ✉e-mail: nasim.sabouri@umu.se

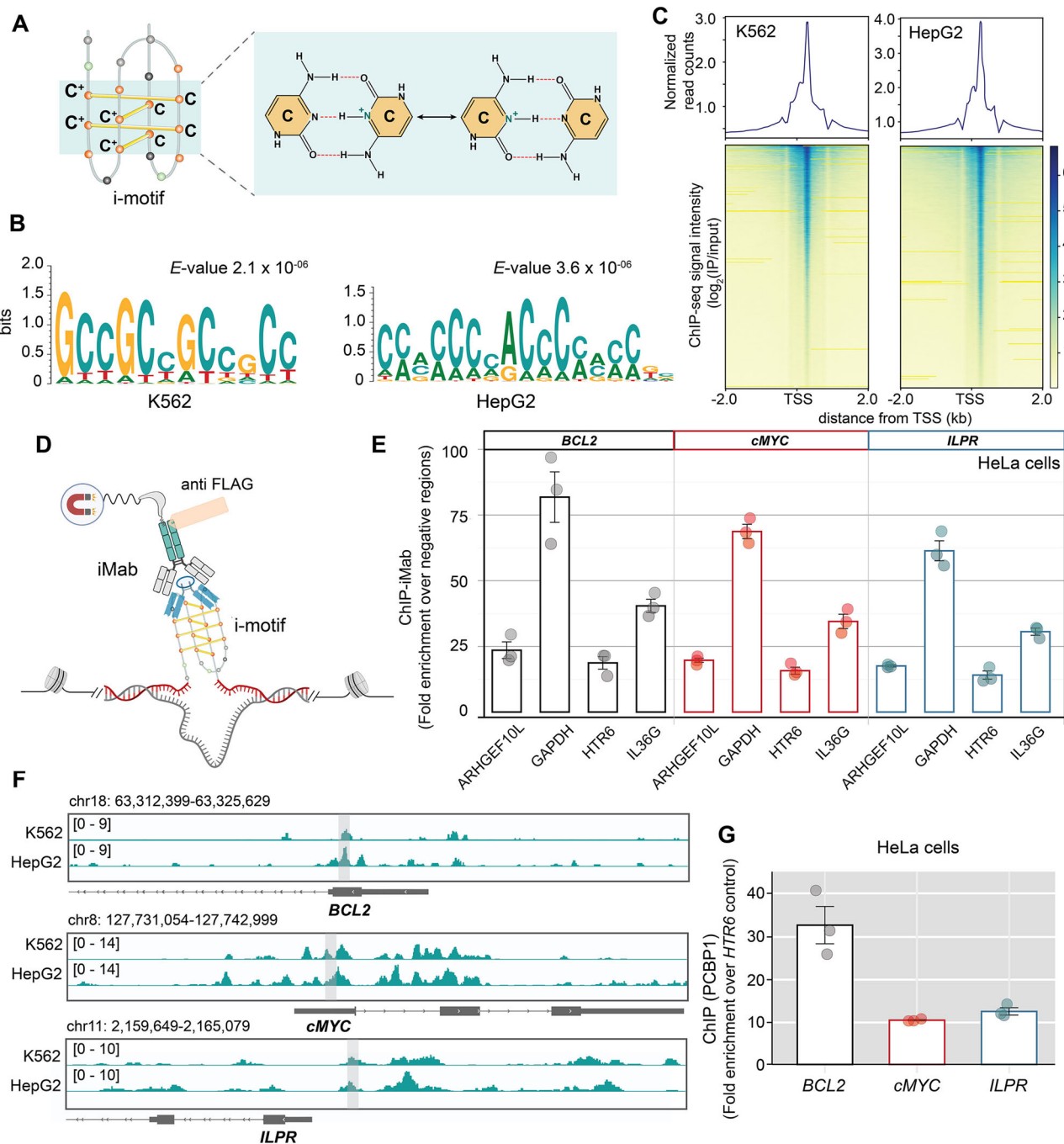

**Fig. 1 | PCBP1 occupancy is enriched at C-rich genomic sequences that potentially form i-motif structures. A** Schematic illustration of i-motif DNA structure, highlighting intercalation of C:CH+ base pairs between two Cs, where one C is in its protonated form (CH+). **B** MEME (Multiple Expectation maximizations for Motif Elicitation)-based motif-enrichment analysis showing significant enrichment of C-rich sequences in PCBP1-bound ChIP-seq peaks from human K562 (GSE174944) and HepG2 (GSE106035) cells; *E*-value indicates statistical significance. **C** Heatmap profile centered around TSS (transcription start site) for PCBP1-ChIP-seq peaks. Read counts were extracted within a region spanning ±2 kb around TSS. The gradient blue-to-yellow color indicates high-to-low counts in the corresponding region. Plots at the top of the heatmaps show the normalized read counts at genomic regions centered at TSS. **D** Schematic diagram of iMab-ChIP, illustrating iMab binding to i-motif regions in crosslinked chromatin, captured by anti-FLAG antibody on magnetic beads. Created in BioRender. Sabouri, N. (2026) https://BioRender.com/84wqzkv. **E** Quantification of iMab-ChIP efficiency at i-motif-

harboring promoters of *cMYC*, *BCL2* oncogenes and insulin gene *ILPR* versus several non-i-motif negative control sites (*ARHGEF10L*, *GAPDH*, *HTR6*, *IL36G*) in HeLa cells. The *y*-axis shows fold enrichment over these negative regions. **F** IGV (Integrative Genomic Viewer) tracks visualizing ChIP-seq data of PCBP1-bound regions at i-motif-containing promoters in K562 and HepG2 cells. Each track shows the signal intensity range, with a scale noted in brackets for each genomic region, indicating the level of PCBP1 occupancy across those genomic regions for different cell types. Highlighted light gray regions overlap with corresponding i-motif-forming sequences under investigation. **G** Quantification of PCBP1-ChIP at three i-motif-containing promoters (*cMYC*, *BCL2*, *ILPR*) compared to the *HTR6* control site in HeLa cells to determine fold enrichment over the *HTR6* site. ChIP-qPCR bar plots in (**E**, **G**) represent the means ± SD of three technical replicates (*n* = 3) and error bars indicate standard deviation (SD). Raw *C*t values from input, IP, and mock (IgG) samples obtained from three independent biological replicates, each measured with three technical replicates, are provided in Supplementary Data 1 and 3.

lack of robust i-motif-detecting tools. In addition, the pH sensitivity of proteins makes interaction studies challenging, as reduced protein stability at lower pH can impede accurate assessment of i-motif binding. Recent proteomic studies uncovered several candidate proteins potentially involved in i-motif binding[19]. Among them, hnRNP (Heterogeneous Nuclear Ribonucleoprotein) or PCBP (Poly-(rC)-binding Protein)-family proteins showed the most promising in vitro interactions with C-rich sequences[20–25]. However, their precise roles in i-motif binding and regulation are underexplored. Immunofluorescence studies demonstrated PCBP1's (hnRNP-E1) co-localization with i-motif foci[26,27]. HnRNP-K has been reported to unfold *cMYC*-i-motif in vitro[21]. While these studies provided indirect evidence of PCBP's potential role in i-motif binding, they did not offer a mechanistic understanding of how PCBP regulates i-motifs at specific genomic loci. This is particularly important in the evolving field of i-motif research, because several independent studies observed i-motif's transient nature at specific loci, wherein the same genomic regions may not consistently fold into i-motifs across various cell types or under different intracellular conditions[15,16,24]. This may cause oversight of specific folding events at particular loci or the dismissal of sporadic observations, necessitating a systematic investigation to characterize PCBP's role in i-motif regulation at specific loci and its cellular implications.

Here, we investigated the multifunctional protein, PCBP1 (Poly(rC)-binding protein 1), which belongs to the hnRNP family[28,29] and is known to bind C-rich RNA/DNA sequences[30,31]. PCBP1 contains three K-homology (KH) domains and plays key roles in oxidative stress response[32], genomic integrity maintenance[33,34], and cell cycle regulation[31,35–37]. While aberrant PCBP1 expression is linked to cell cycle arrest in various cancers[31,37], its molecular mechanisms remain unclear. Our study illustrates the mechanism of i-motif regulation by different domains of PCBP1 and establishes its implications to regulate $G_1/S$ transition during the cell cycle. Using chromatin immunoprecipitation (ChIP) and cell-based studies, we demonstrated that PCBP1 is enriched at specific i-motif foci in the genome in a cell cycle-dependent manner, allowing these i-motifs to transiently accumulate during $G_1/S$ transition and unfold in early S phase to facilitate replication. PCBP1 depletion caused $G_1/S$ arrest and increased replication stress, reinforcing that PCBP1-mediated i-motif unfolding is essential for proper $G_1/S$ transition. We also illustrated the molecular basis of specific interactions between PCBP1 and i-motifs by integrating spectroscopic studies, molecular dynamics (MD) simulations, and biochemical assays, which collectively revealed that PCBP1 selectively unfolds specific i-motifs with distinct kinetics, depending on their protonation states and hairpin-forming tendencies. We demonstrated a hierarchical mechanism for i-motif resolution, where the cooperative action of KH1 + 2 domains remodels i-motifs, priming them for KH3-driven unfolding. Our findings lay the foundation for future studies exploring how PCBP1 orchestrates i-motif dynamics to ensure genomic stability and fidelity of $G_1/S$ transition.

## Results

### PCBP1 is enriched at C-rich, potential i-motif-forming genomic sites

PCBP1 is well known for its affinity toward C-rich DNA sequences[31,38]. However, its specific role in i-motif formation and regulation remains unclear. To investigate PCBP1's genomic binding preferences, we performed MEME-based motif-enrichment analysis[39] on published PCBP1-ChIP-seq datasets (Fig. 1B), which identified significantly enriched C-rich motifs with interspaced 3C/2C-repeats, resembling consensus iMfs (Fig. 1A, B). A global analysis of ChIP-seq signals further revealed significant enrichment of PCBP1-peaks around ±2 kb of TSS (transcription start site) (Fig. 1C) overlapping with several promoters, including *cMYC, BCL2, ILPR, PDGFa, VEGFa*, and *HIF1α* (Fig. 1F, Supplementary Fig. 1A) that harbor C-rich sequences capable of forming i-motifs in vitro[40–43]. To assess if these promoter sequences form

i-motifs in cells, we performed ChIP-quantitative PCR (ChIP-qPCR) with iMab antibody. While iMab is widely regarded for its selective i-motif binding within cells[11], recent studies argued over its potential interactions with non-i-motif-forming-C-rich sequences[44]. To address this, we compared iMab occupancy in the target regions to several control sites (Fig. 1D) with moderate C-richness (~50%) (Supplementary Table 2), but no evidence of i-motif formation based on prior G4-ChIP-seq[45] and iMab-Cut-and-Tag-seq datasets[15]. We detected no significant immunoprecipitation (IP)-iMab signal at *HIF1α* promoter, while positive IP-iMab signals for *VEGFa* and *PDGFa*, albeit only against two of the four C-rich control regions (Supplementary Fig. 1B). For *cMYC, BCL2*, and *ILPR* promoters, we observed ~10-fold higher iMab occupancy compared to all control sites (Fig. 1E), suggesting their strong potential to form i-motifs in HeLa cells. Subsequently, we validated PCBP1's interactions with these sequences using ChIP-qPCR with PCBP1-specific antibody, which demonstrated robust PCBP1 occupancy at these sites (Fig. 1G). Collectively, our findings indicate that PCBP1 preferentially binds to C-rich genomic sequences that potentially form i-motifs.

### PCBP1 selectively binds i-motifs over their unfolded forms in vitro

Since PCBP1 preferentially interacts with i-motif-forming genomic regions, we hypothesized that PCBP1 would bind iMfs in vitro. As both proteins and i-motifs' stability are pH-dependent, we first investigated their pH tolerance. At pH 6.4, 7.0, or 8.0, purified PCBP1 showed the characteristic double-dip α-helix structure in circular dichroism (CD) spectra[46] (Supplementary Fig. 2A, B). At pH 6.0 and 5.5, the negative peak around 210–220 nm became less pronounced, suggesting partial structural changes (Supplementary Fig. 2B). This was supported by thermal-shift assays, which revealed about 5 °C decrease in PCBP1's melting temperature ($T_m$) at pH 6.0 or 5.5, compared to pH 6.4 and higher (Fig. 2A, Supplementary Table 3), suggesting that PCBP1 maintains optimal functions and structural integrity at ≥ pH 6.4. Next, we obtained CD spectra of *cMYC, BCL2, ILPR*, hTeloC (C-rich telomeric strand)-iMfs that yielded robust iMab-positive signals to evaluate pH-dependent stability and transition pH ($pH_T$) (where 50% of i-motifs remains folded at a particular pH) (Fig. 2B, Supplementary Fig. 3). All sequences demonstrated intramolecular i-motif spectra with a signature CD profile of positive and negative maxima around 287 and 260 nm, respectively[40], and $pH_T$ values above pH 6.4 (Fig. 2B, Supplementary Figs. 3 and 4). These findings allowed us to investigate PCBP1 binding to iMfs at pH 6.4 and 8.0, ensuring PCBP1's stability. At pH 6.4, a significant proportion of i-motifs remained folded, whereas at pH 8.0, they were unfolded, allowing comparison of PCBP1's affinity to both structural states of the same sequence.

The electrophoretic mobility shift assays (EMSAs) at pH 6.4 and 8.0 revealed that PCBP1 binds to both i-motifs and their unfolded states; nevertheless, complete band-shifts occurred at lower PCBP1 concentrations at pH 6.4, indicating PCBP1's stronger preference for i-motifs (Fig. 2C, D). Notably, EMSA gels revealed distinct migration patterns of PCBP1–iMfs complexes between pH 6.4 and 8.0. At pH 8.0, only a single-shifted band was detected. However, at pH 6.4, two slow-migrating shifted complexes appeared for all iMfs, most prominently for *BCL2*, that increased in intensity with rising PCBP1 concentrations. These multiple bands at pH 6.4 likely arise from i-motif's structural heterogeneity induced by PCBP1 interactions or transient PCBP1 multimer formation. To clarify this, we performed ITC (Isothermal titration calorimetry) (Supplementary Fig. 5), which fitted into one-site binding models at both pH levels, thereby excluding higher-order binding stoichiometries.

To quantify binding affinities, we performed microscale thermophoresis (MST). Because subtle pH changes between pH 6.4 and 8.0 may influence both i-motif equilibrium and protein charge distribution, we also measured affinities at pH 7.0 as an intermediate condition (Fig. 2E, Supplementary Fig. 6A–E). To further evaluate PCBP1's selectivity towards i-motifs, we tested G4-forming or non-GC-rich

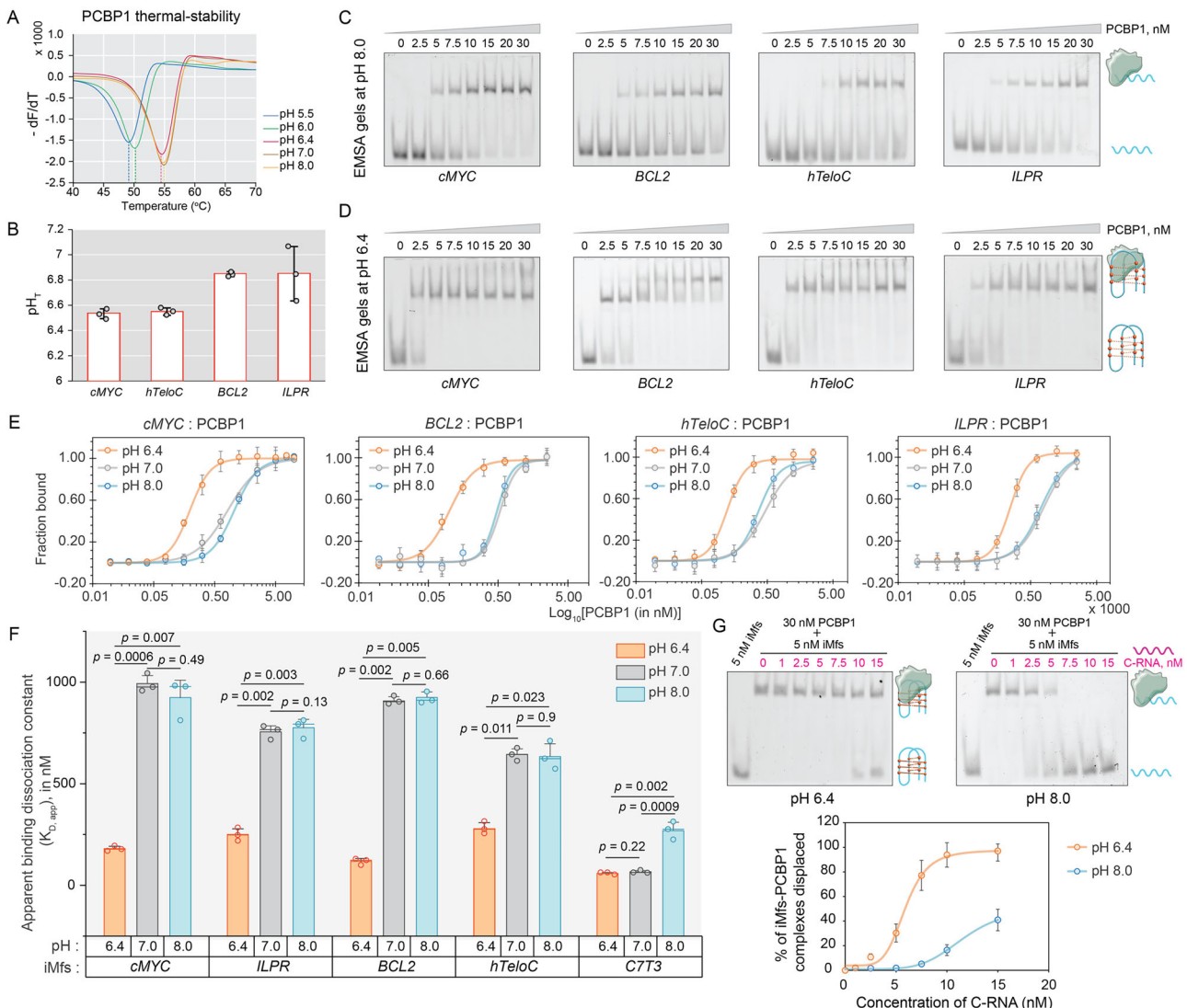

**Fig. 2 | PCBP1 exhibits higher affinity for i-motif DNA compared to its unfolded states in vitro. A** Thermal-shift assay of PCBP1 at different pH levels. The $y$-axis shows $-\frac{dF}{dT}$ representing the negative derivative of Sypro-Orange fluorescence signal with respect to temperature; melting temperature ($T_m$) defined by negative maxima at different pH values. **B** $pH_T$ values indicate the transition pH, at which 50% of i-motifs remain folded at 25 °C, derived from CD spectroscopy for *cMYC, BCL2, ILPR*, and hTeloC-i-motifs. **C** EMSA gels at pH 8.0 and **D** at pH 6.4, showing 5′-Cy5-labeled iMfs with band-shifts upon incubation with increasing concentrations of recombinantly purified PCBP1. **E** Sigmoidal curves from MST binding experiments between

PCBP1 and iMfs at pH 8.0, 7.0, and 6.4. **F** $K_{D,app}$ estimated from MST binding curves for iMfs at pH 8.0, 7.0, and 6.4. Statistical significance and $p$ value calculated using two-tailed Student's $t$-test, where $p$ values < 0.05 are considered statistically significant. **G** Competitive EMSA showing displacement of iMfs-PCBP1 complexes with increasing concentration of C-RNA performed at pH 6.4 and 8.0. Native gels (top) and densitometric analyses of gels (bottom). Data in (**B, E, F, G**) represent the mean ± SD from three independent biological replicates ($n = 3$). Error bars indicate standard deviation. Uncropped and unprocessed gel images corresponding to EMSA experiments shown in (**C, D, G**) are provided in the Source data.

sequences, which showed significantly weaker binding than i-motifs (Supplementary Fig. 7, Supplementary Table 4). Consistent with EMSA, PCBP1 showed >2-fold higher affinity for i-motifs at pH 6.4 than their unfolded counterparts at pH 7.0 or 8.0, with apparent dissociation constants ($K_{D,app}$) ranging between 150 and 300 nM (Fig. 2F, Supplementary Table 4). To verify whether PCBP1's selectivity depends on structural folding rather than sequence composition, we included a synthetic and well-characterized C7T3-i-motif[47,48] that remains stable at pH 7.0 due to its high C-density. For promoter and hTeloC iMfs, which unfold at pH 7.0 and higher, PCBP1 binding showed no significant difference between pH 7.0 and 8.0. However, C7T3, which forms a stable i-motif at pH 6.4 and 7.0 but unfolds at pH 8.0, displayed stronger $K_{D,app}$ values at pH 6.4 and 7.0 than at pH 8.0 (Fig. 2F, Supplementary Fig. 8), reinforcing PCBP1's binding preference for i-motifs over unfolded forms. Notably, C7T3's unfolded form at pH 8.0 still

bound to PCBP1 with sub-micromolar binding affinity, which is comparable to other i-motifs at pH 6.4, suggesting that both folded and unfolded states may contribute to PCBP1's biological interactions, influenced by i-motif stability and C-density of the sequence. To probe the thermodynamic basis of this preference, we performed ITC, revealing a free energy difference ($\Delta\Delta G$) of 0.5–0.6 kcal mol$^{-1}$ between pH 6.4 and pH 8.0 (Supplementary Fig. 9). This corresponded to ~two-fold higher affinity at pH 6.4 (i-motif folded state), consistent with MST findings.

We further assessed binding selectivity by competitive EMSA using a canonical C-rich RNA (C-RNA) substrate (Supplementary Table 1) of PCBP1[31]. At pH 8.0, equimolar RNA concentrations displaced ~50% of the pre-formed PCBP1–iMfs complex with complete displacement at a two-fold RNA excess (Fig. 2G). In contrast, at pH 6.4, even a three-fold RNA excess displaced only ~40%, indicating that

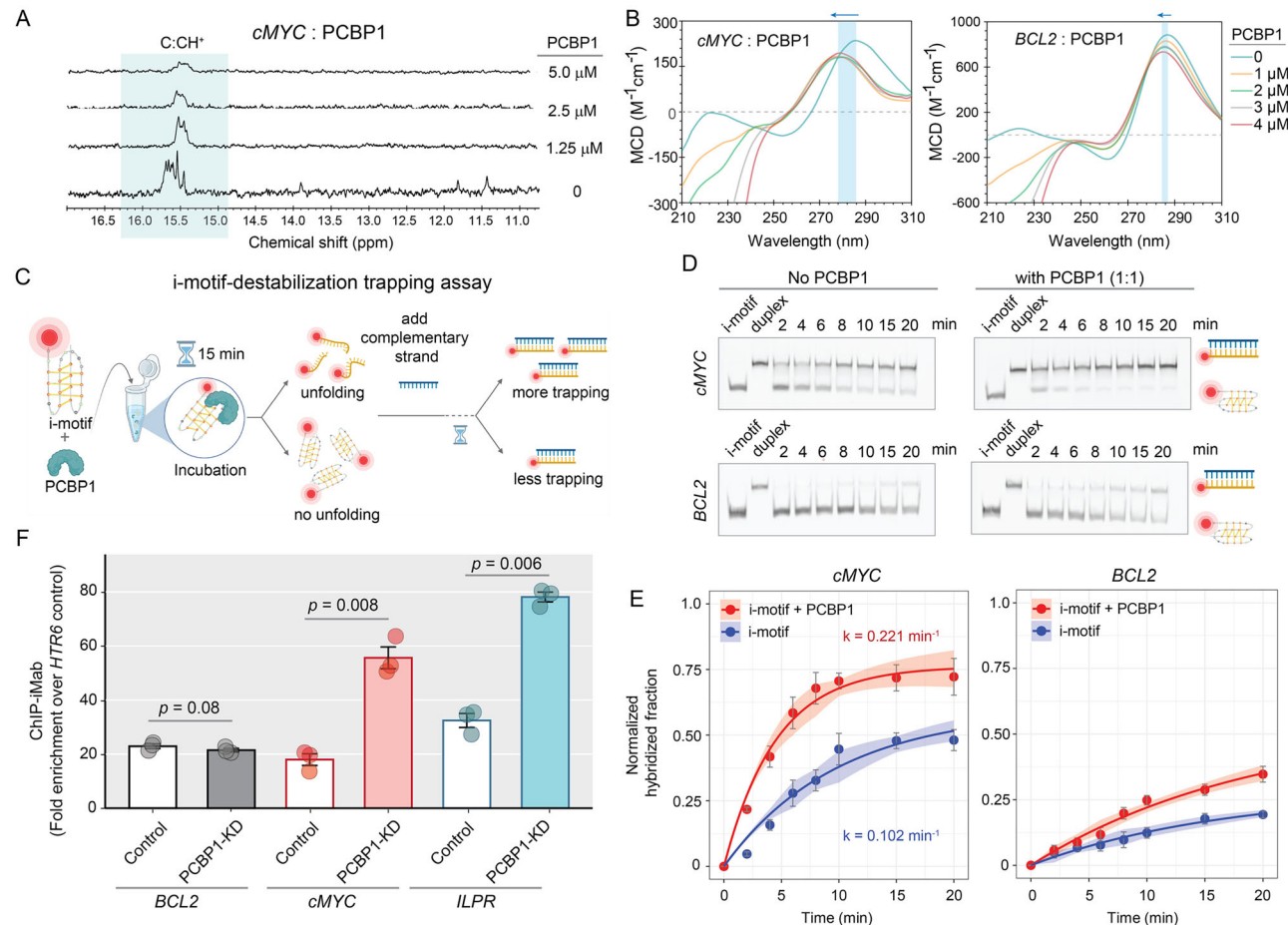

**Fig. 3 | PCBP1 unfolds *cMYC*-i-motifs with faster kinetics than *BCL2*-i-motifs in vitro. A** [1]H NMR spectra. Broadening of imino proton resonances (15–16 ppm defining C:CH[+] bonds) of *cMYC*-i-motif during PCBP1 titrations. **B** CD spectra of *cMYC* and *BCL2*-i-motifs upon PCBP1 titrations, with highlighted regions showing blue-shift (with arrows) and hypochromic effects. **C** Schematic diagram of i-motif-destabilization trapping assay. PCBP1 and i-motifs are incubated for 15 min, followed by the addition of complementary strands or traps; reactions were stopped at various time points until 20 min. PCBP1-driven i-motif unfolding led to increased hybridization rates (*k*) between i-motifs and their corresponding complementary strands. Created in BioRender. Sabouri, N. (2026) https://BioRender.com/oazozsm. **D** Native PAGE visualizing aliquots of hybridization reactions at different time intervals upto 20 min, alongside negative control (i-motifs without complementary strands) and positive controls (hybridized duplex for 24 h) in the absence and presence of PCBP1 in 1:1 molar ratio. Uncropped and unprocessed gel images

corresponding to native PAGE experiments are provided in the Source data. Created in BioRender. Sabouri, N. (2026) https://BioRender.com/oazozsm. **E** Densitometric analysis of i-motif-trapping with and without PCBP1. Normalized intensities of hybridized fractions calculated from three independent biological replicates (*n* = 3) at different time points, fitted to single-exponential function equations. Error bands/bars denote standard deviation. **F** Fold enrichment of iMab-ChIP at i-motif-harboring promoters (*cMYC, BCL2, ILPR*) versus non-i-motif control site (*HTR6*); error bars from standard deviation of three technical replicates (*n* = 3). Statistical differences and *p* values between control and PCBP-KD determined by one-way ANOVA followed by Tukey–Kramer post hoc test, where *p* values < 0.05, are considered statistically significant. Raw $C_t$ values from input, IP, and mock (IgG) samples obtained from three independent biological replicates, each measured with three technical replicates, are provided in Supplementary Data 2.

unfolded i-motifs form weaker complexes with PCBP1 that are readily outcompeted by C-rich RNA, whereas folded i-motifs provide a more stable and preferred binding platform for PCBP1. To address the possibility that PCBP1 binding arises from non-specific electrostatic interactions with protonated Cs, we evaluated a previously studied control C-rich sequence (mutC)[44] that does not fold into an i-motif under our acidic experimental conditions despite its C-richness (Supplementary Fig. 10A, Supplementary Table 1). EMSA at both pH 6.4 and pH 8.0 showed no detectable complex between mutC and PCBP1 (Supplementary Fig. 10B). Together, these data demonstrate that PCBP1 binding is structure-specific and not driven by general electrostatic interactions, and it directly interacts with i-motifs in vitro with higher affinities than their unfolded C-rich forms.

## PCBP1 unfolds different i-motifs with varied kinetics
Having established PCBP1's preferential recognition for i-motifs, we sought to examine how its binding influences i-motif structures. To

directly probe the conformational effects of PCBP1 binding on i-motifs, we performed [1]H-NMR titrations of PCBP1 into i-motifs. We observed significant broadening and up-field shifts in C:CH[+] resonances around 15–16 ppm, suggesting altered i-motif dynamics upon PCBP1 binding (Fig. 3A, Supplementary Fig. 11A–C). However, since spectrum broadening results from both strong binding and protein-induced conformational alterations, we acquired CD spectra of *cMYC*, hTeloC, and *ILPR*-i-motifs, which exhibited hypochromism and distinct blue-shifts upon PCBP1 titrations, suggesting PCBP1-induced structural destabilization of these i-motifs (Fig. 3B, Supplementary Fig. 12A, B). In contrast, *BCL2*-i-motif displayed hypochromism but lacked the pronounced blue-shift (Fig. 3B) in CD, suggesting a variable impact of PCBP1 binding on *BCL2*-i-motif. To quantify the unfolding kinetics, we designed an i-motif-destabilizing-trapping assay (Fig. 3C), where we added i-motifs' complementary strands as traps to hybridize with unfolded i-motifs, preventing their refolding. We hypothesized that if PCBP1 accelerates the unfolding of specific i-motifs, it would

accumulate more hybridized products within a shorter incubation period and faster hybridization rates ($k$) (Fig. 3C). In the absence of PCBP1, unfolding was generally slow, but *BCL2*-i-motif was strikingly slower compared to other iMfs. PCBP1 addition significantly accelerated their unfolding, with varying $k$ across four i-motifs (Fig. 3D, E). Consistent with CD, *cMYC* and hTeloC-i-motifs unfolded most rapidly, *ILPR*-i-motif showed an intermediate rate, while *BCL2*-i-motif remained comparatively resistant to PCBP1-induced destabilization, showing only a modest increase in trap formation (Fig. 3D, E, Supplementary Fig. 13A, B). These results indicate that *BCL2* adopts an intrinsically kinetically persistent fold that impedes spontaneous unfolding, and that this intrinsic stability limits the ability of PCBP1 to promote its unfolding. Next, we examined the binding preferences within cells and performed iMab-ChIP-qPCR following siRNA-mediated knockdown of endogenous PCBP1 (PCBP1-KD) within HeLa cells (Supplementary Fig. 14). Aligning with in vitro observations, PCBP1-KD enhanced iMab occupancy by 3-fold at *cMYC* and *ILPR* promoters, suggesting more stable i-motif formation at these regions upon PCBP1-KD (Fig. 3F, Supplementary Fig. 15). However, no significant change in iMab occupancy was observed at the *BCL2* promoter, reinforcing that PCBP1 differentially regulates i-motif stability in a sequence-dependent manner. Together, these results reveal that while PCBP1 preferentially recognizes i-motif structures, its binding exerts variable effects on their conformational stability and unfolding kinetics, likely reflecting sequence-intrinsic structural features within individual iMfs.

## PCBP1-mediated i-motif-unfolding depends on their protonation and hairpin-forming potential

Since PCBP1 resolves different i-motifs at varying rates, we investigated why certain i-motifs resist PCBP1-mediated unfolding. We hypothesized that i-motif's intrinsic stability could be a contributing factor. To test this, we examined C7T3-i-motif, which has a $pH_T$ of 7.1[49], and is more stable at pH 6.4 than other i-motifs tested. CD titrations revealed stronger blue-shifts and hypochromism in C7T3 (Supplementary Fig. 16), compared to *BCL2*, though less pronounced than *cMYC* or hTeloC, and similar to *ILPR*, suggesting efficient but variable PCBP1-driven unfolding. C7T3's greater protonation state, due to its high C-richness at pH 6.4, may contribute to its distinct unfolding. However, *BCL2*, despite having similar $pH_T$ as *ILPR*, showed negligible unfolding, indicating that factors beyond protonation, such as structural features, may influence its resistance to PCBP1-mediated unfolding.

Since i-motifs often co-exist with hairpin structures, their unfolding may facilitate hairpin formation[22,42,50]. To explore this, we examined *BCL2*-i-motif using [1]H NMR, which revealed proton resonances between 12–14 ppm, indicating Watson-Crick base-pairing in hairpin structures (Supplementary Fig. 11A), consistent with earlier studies[42]. To determine if PCBP1-mediated i-motif-unfolding is influenced by hairpin formation, we performed high-resolution primer extension assays[51] with *cMYC* and *BCL2*-i-motifs, and non-i-motif controls, with or without PCBP1 (Fig. 4A–D, F, Supplementary Figs. 17 and 18). In this assay, DNA polymerase stalls at replication obstacles (e.g., i-motif or hairpin), producing distinct bands on denaturing gels. To enhance i-motif stability under the conditions dictated by the replication assay buffer, we conducted these assays at a more acidic pH (6.0 instead of 6.4), where PCBP1 undergoes only a mild structural rearrangement compared to pH 6.4 (Supplementary Fig. S19). No pausing signal was detected in the non-i-motif control template (Supplementary Fig. 17), while in *cMYC*, we observed a single stalling/pausing signal one-nucleotide preceding i-motif, suggesting that *cMYC*-i-motif acts as a stable replication barrier. PCBP1-binding significantly enhanced replication through *cMYC*, yielding ~1.5× more full-length products, suggesting that efficient i-motif-unfolding by PCBP1 facilitates replication (Fig. 4B–D). Conversely, *BCL2* exhibited three stalling/pausing signals at C2, G6, and G10 (Fig. 4B, F–H), with fewer full-length products compared to *cMYC*. Although PCBP1 slightly

increased the full-length product in *BCL2*, it was less efficient at unfolding *BCL2*-i-motif (Fig. 4F). While the pausing preceding C2 attributes to i-motif formation, we investigated whether other pausing signals were due to hairpin formations that inhibit PCBP1-driven unfolding. We designed two *BCL2* mutants: G10TG12T and $G_{all} \rightarrow T$ (all Gs mutated to Ts) that disrupt hairpin formation. [1]H NMR confirmed no Watson-Crick-associated proton resonances at 12–14 ppm in these mutants (Fig. 4E). Primer extension assays showed reduced G10-associated stalling in both mutants and ~1.5× more full-length products with PCBP1, indicating that G10 contributes to hairpin formation and acts as a structural barrier to PCBP1-driven unfolding. Nevertheless, stalling at G6, before the second C-tract, was unaffected and may represent another i-motif because *BCL2*-i-motifs form dynamic and heterogeneous conformers involving different C-tract combinations, similar to polymorphic G4s (Fig. 4F–H)[24,42,52]. These findings were validated by CD, indicating stronger blue-shifts and hypochromism in *BCL2* mutants upon PCBP1 titrations than wild-type sequence (Figs. 3B and 4I). Therefore, PCBP1-driven unfolding depends on i-motifs' protonation states, while sequences with co-existing hairpins significantly hinder the unfolding (Fig. 4J).

## PCBP1 depletes i-motif formation at specific loci during the cell cycle

Functional enrichment analysis of a recent proteomic study linked candidate i-motif-binding proteins to cell cycle regulation, which complemented previous observations of cell cycle-specific spatio-temporal changes in global i-motif foci[8,10,19]. PCBP1 is known from earlier studies to control cell cycle[31,35–37,53,54], and is now shown by us to regulate i-motif dynamics within cells. To explore this connection, we examined the impact of PCBP1-KD on cell cycle progression. Consistent with previous reports[37], PCBP1-KD caused $G_1$/S arrest (Fig. 5A, B). This was substantiated by dot-blot assays, which showed minimal BrdU (5-bromo-2'-deoxyuridine) incorporation even 4 h after the synchronized $G_1$/S cells were released upon PCBP1-KD, indicating a failure to initiate replication (Fig. 5C). In parallel, PCBP1-KD cells showed elevated γH2AX levels (Fig. 5D), suggesting DNA damage accumulation, which potentially contributed to $G_1$/S arrest as a protective checkpoint response, allowing time for DNA repair before S phase entry. To determine whether PCBP1's roles in cell cycle control and i-motif regulation are linked, we synchronized cells using a double-thymidine block at $G_1$/S or S phase (Supplementary Fig. 20A, B) and analyzed PCBP1 occupancy at three oncogenic i-motif loci by PCBP1-ChIP-qPCR. PCBP1 occupancy was significantly higher at *cMYC* and *ILPR* in $G_1$/S, and decreased in S phase compared to asynchronous cells, while its occupancy at *BCL2* remained unchanged (Fig. 5E, F). To examine whether i-motif formation at these loci followed a similar pattern, we performed iMab-ChIP-qPCR in synchronized cells. Except for *BCL2*, iMab occupancy was higher at *cMYC* and *ILPR* during $G_1$/S, and decreased in early S phase (Fig. 5G). Since i-motif formation influences G4 dynamics in the complementary strands[55], we also examined BG4 occupancy to detect G4 formation[56], revealing pronounced BG4 occupancy at all three promoters compared to non-G4 control sites during both $G_1$/S and S phases. However, at *cMYC*, BG4 occupancy was significantly increased during S phase, compared to $G_1$/S transition (Supplementary Fig. 21A), aligning with earlier reports on the cell cycle-dependent dynamics of G4 structures[56,57]. In PCBP1-KD cells, BG4 occupancy at *cMYC* was significantly reduced, while *BCL2* and *ILPR* remained unchanged (Supplementary Fig. 21B). This likely reflects the reduced proportion of S phase cells upon PCBP1-KD (Fig. 5A, B), rather than a direct regulatory role of PCBP1 on G4s, considering its weak in vitro binding to G4s (Supplementary Fig. 7). In contrast, the temporal dynamics of i-motif formation at *cMYC* and *ILPR* loci during the $G_1$/S and S phases closely mirrored PCBP1 occupancy, suggesting a coordinated regulation. To further assess the functional consequences of PCBP1-mediated resolution of specific i-motifs, we

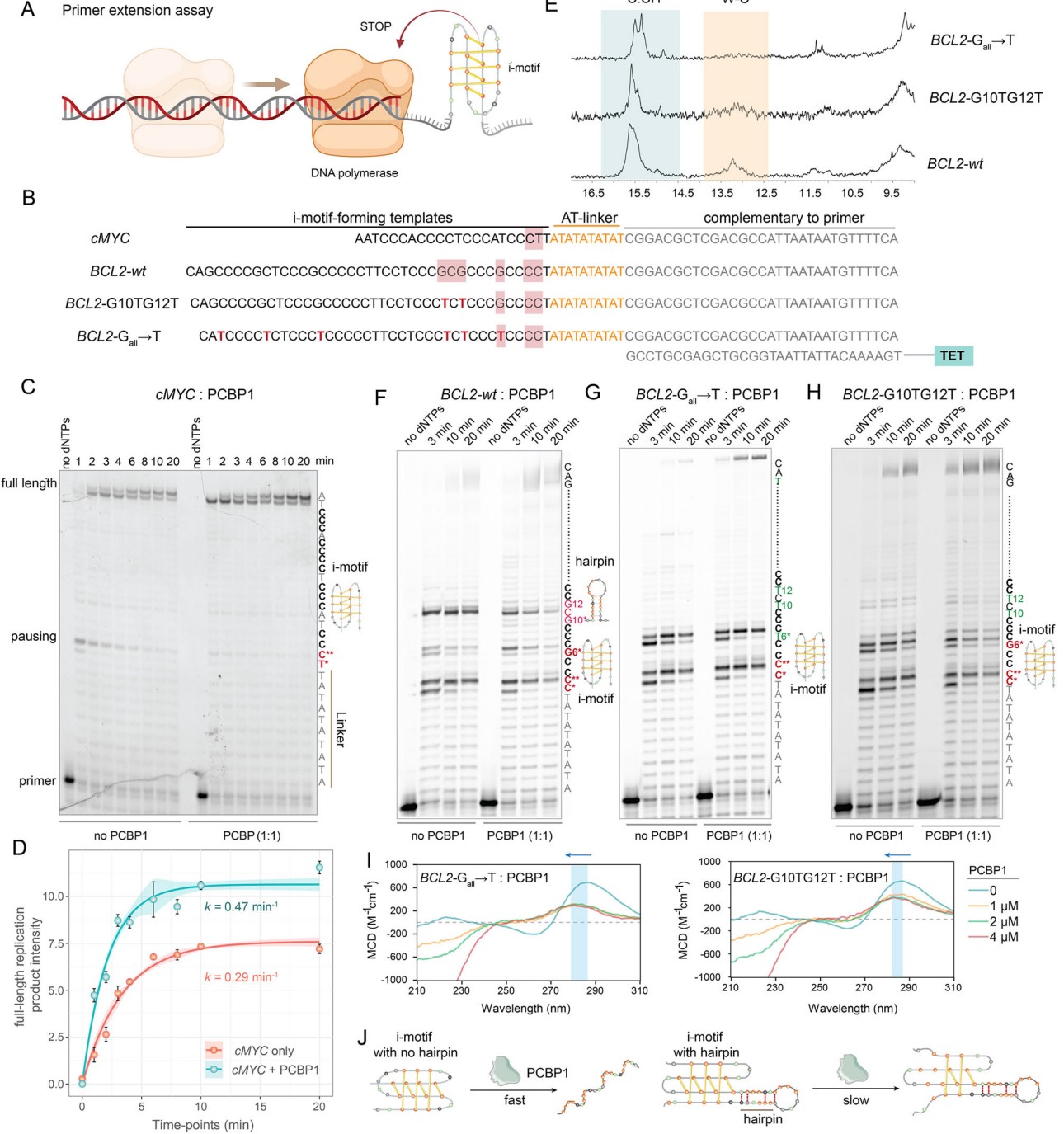

**Fig. 4 | I-motifs co-existing with hairpin structures resist PCBP1-mediated unfolding. A** Schematic of primer extension assay on i-motif-forming templates, where i-motif formation sterically blocks or slows DNA polymerase progression. Created in BioRender. Sabouri, N. (2026) https://BioRender.com/vwd8yby. **B** Primer extension templates containing i-motif-forming sequences, AT-linker, and TET (tetrachlorofluorescein)-labeled primer complementary sequences. G-to-T mutations are marked in red. Pausing signals are highlighted in pink boxes. **C** Primer extension assay with *cMYC*-i-motif-forming template at various time points until 20 min. AT-linker preceding *cMYC*-i-motif and sequence provided with pausing signals (red, asterisk), full-length products, and primer. **D** Densitometric analyses of full-length replication product formation over time in the presence of PCBP1; results expressed as means ± SD from three biological replicates (*n* = 3), fitted to single-exponential function equations to calculate the rate of reaction (*k*). Error bars/bands denote standard deviation. **E** Imino proton NMR resonances (15–16 ppm) and Watson-Crick (W-C) pairing (12–14 ppm) of wild-type and mutated

*BCL2*-i-motifs. **F** Primer extension assay with wild-type *BCL2*-i-motif-forming template (*BCL2*-wt) at different time points. AT-linker preceding *BCL2*-wt-i-motif, the template sequence with three pausing signals highlighted with red asterisks; full-length product and primer are shown with potential involvement of G10 and G12 in hairpin formation that blocks PCBP1. Primer extension assays involving **F** *BCL2*-wt, **G** *BCL2*-G$_{all}$ → T, or **H** *BCL2*-G10TG12T i-motif-forming templates at different time points. Gels shown in (**F**–**H**) are representative of three independent biological replicates. Mutations at specific Gs (marked in green) to prevent hairpin formation; pause signals for i-motif formation marked in red and asterisks. **I** CD titrations with increasing PCBP1 concentrations on *BCL2*-G$_{all}$ → T and *BCL2*-G10TG12T; blue-shifts and hypochromic effects indicated by blue bars and arrows. **J** Schematic illustration showing hairpin-forming potential of i-motif significantly slows down PCBP1's ability to unfold i-motifs. Uncropped and unprocessed gel images corresponding to primer extension assays shown in (**C**, **F**, **G**, **H**) are provided in the Source data.

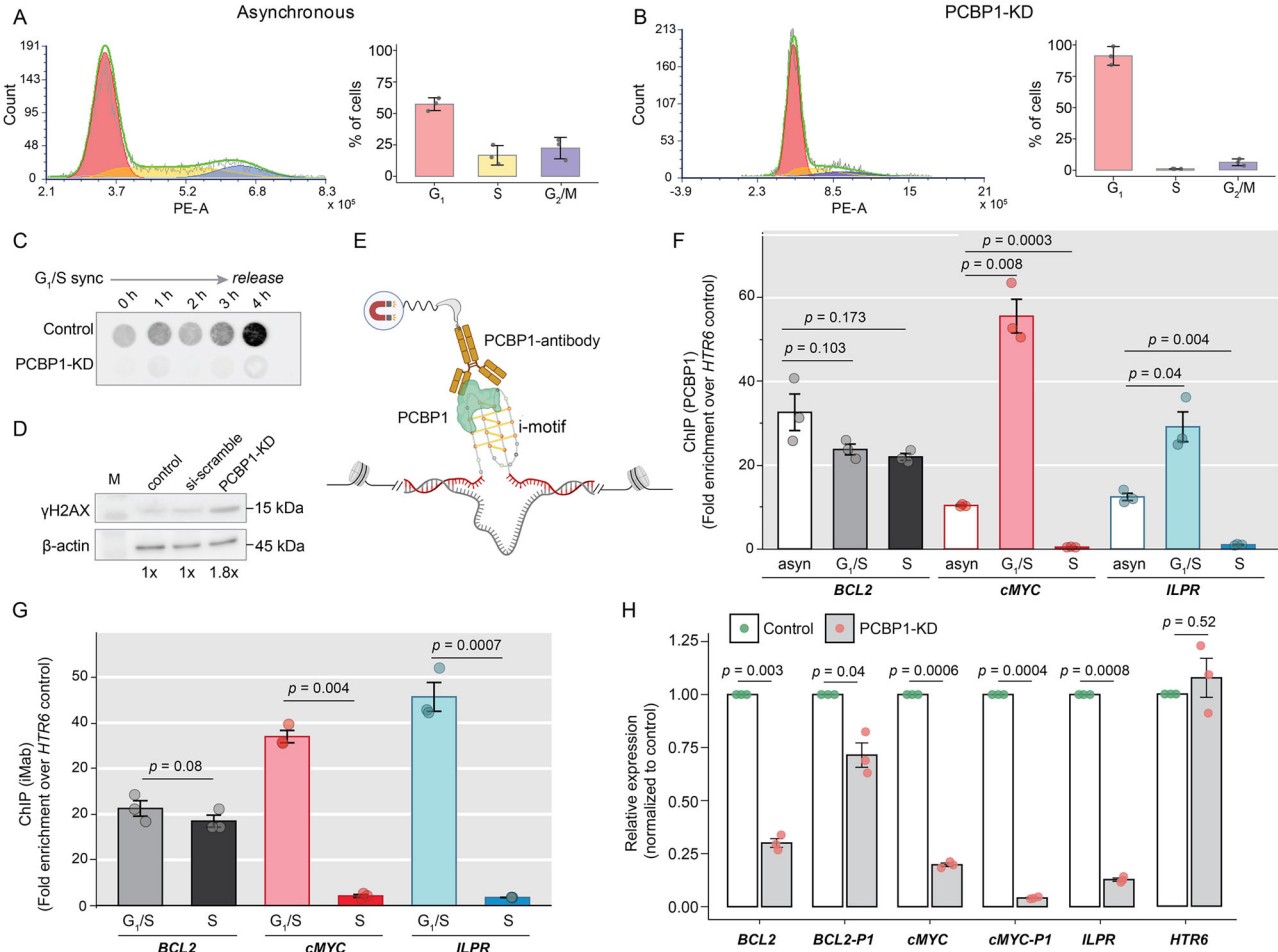

**Fig. 5 | PCBP1 regulates i-motif folding in HeLa cells in a cell cycle-dependent manner.** Flow cytometry-based analysis of DNA content and cell cycle distribution in **A** asynchronous and **B** PCBP1-KD-treated HeLa cells; bar plots showing cell distribution in $G_1$, S, and $G_2$/M phases. PE-A Phycoerythrin–Area. Results expressed as means ± SD from three biological replicates ($n = 3$). **C** Anti-BrdU dot-blot assay on $G_1$/S-synchronized genomic DNA, followed by synchrony release at different time points upto 4 h in control and PCBP1-KD-treated HeLa cells. **D** Western blot of γH2AX upon 48 h of PCBP1-KD. Si-scrambled siRNA as negative control; β-actin as housekeeping protein. Fold-change increase in γH2AX below the blot. Blots shown are representative of three independent biological replicates. Uncropped and unprocessed images corresponding to dot-blot and western blot experiments shown in (**C**, **D**) are provided in the Source data. **E** Schematic illustration of PCBP1-ChIP using PCBP1-specific antibody. Created in BioRender. Sabouri, N. (2026) https://BioRender.com/uvavqla. **F** Fold enrichment of PCBP1-ChIP at i-motif-

containing promoters compared to *HTR6* control site in asynchronous (asyn), $G_1$/S- and S-synchronized HeLa cells. **G** Fold enrichment of iMab-ChIP at i-motif-harboring promoters (*cMYC*, *BCL2*, *ILPR*) over non-i-motif control site (*HTR6*) in asynchronous, $G_1$/S- and S-synchronized HeLa cells. ChIP bar plots and error bars in (**F**, **G**) represented as means ± SD of three technical replicates ($n = 3$). Statistical differences and *p* values determined by one-way ANOVA followed by Tukey–Kramer post hoc test, where *p* values < 0.05 are considered statistically significant. Raw $C_t$ values from input, IP, and mock (IgG) samples obtained from three independent biological replicates, each measured with three technical replicates, are provided in Supplementary Data 3 and 4. **H** Relative expression ($2^{-\Delta\Delta Ct}$) in i-motif-containing and non-i-motif-containing gene expression upon PCBP1-KD using RT-PCR. Data and error bars represent the mean ± SD from three independent biological replicates ($n = 3$). *P* values calculated using two-tailed Student's *t* tests, where *p* values < 0.05 are considered statistically significant.

examined their transcript levels upon PCBP1-KD. Because promoter activity at *cMYC* and *BCL2* is influenced by G4/i-motif structural transitions at their P1 promoters[58,59], we specifically quantified P1-derived transcripts by RT-qPCR. We observed reduced transcript levels in all i-motif-containing genes upon PCBP1-KD, despite PCBP1's selectivity toward i-motif unfolding (Fig. 5H). This likely reflects a combination of direct effects, stemming from impaired i-motif resolution at these promoters, and indirect consequences of PCBP1-KD, given PCBP1's multifunctional nature and the involvement of additional regulatory proteins at these loci (e.g., Nucleolin, hnRNP-K, hnRNP-LL)[21,22,60]. Collectively, our findings indicate that PCBP1 facilitates unfolding of specific i-motifs in a cell cycle-dependent manner and that disruption of this regulation not only alters the temporal dynamics of i-motif formation but also impacts transcriptional output from these loci, likely through a combination of direct structural effects and broader PCBP1-dependent regulatory pathways.

## Role of PCBP1-KH domains in i-motif-binding and unfolding in vitro

Although we show that PCBP1 unfolds i-motifs with varied kinetics, its molecular mechanism remains unclear. To address this, we truncated PCBP1 into KH domains–critical for DNA binding and examined their contributions in i-motif selectivity and unfolding[36,61,62]. Previous research on hnRNP-K (a PCBP1-family protein) demonstrated that individual KH domains unfold *cMYC*-i-motif, albeit less efficiently than the full-length protein[21]. Considering the distinct nucleic acid binding behaviors and amino acid compositions of PCBP proteins (Supplementary Figs. 29 and 30A, B), we purified PCBP1's KH1 + 2 and KH3 domains (Fig. 6A, Supplementary Fig. 22A–C). Unfortunately, due to precipitation, likely caused by a long unstructured region between KH2 and KH3, KH2 + KH3 purification was unsuccessful.

We used thermal-shift assays (Supplementary Fig. 23A, B, Supplementary Table 5) to assess the pH tolerance of KH mutants,

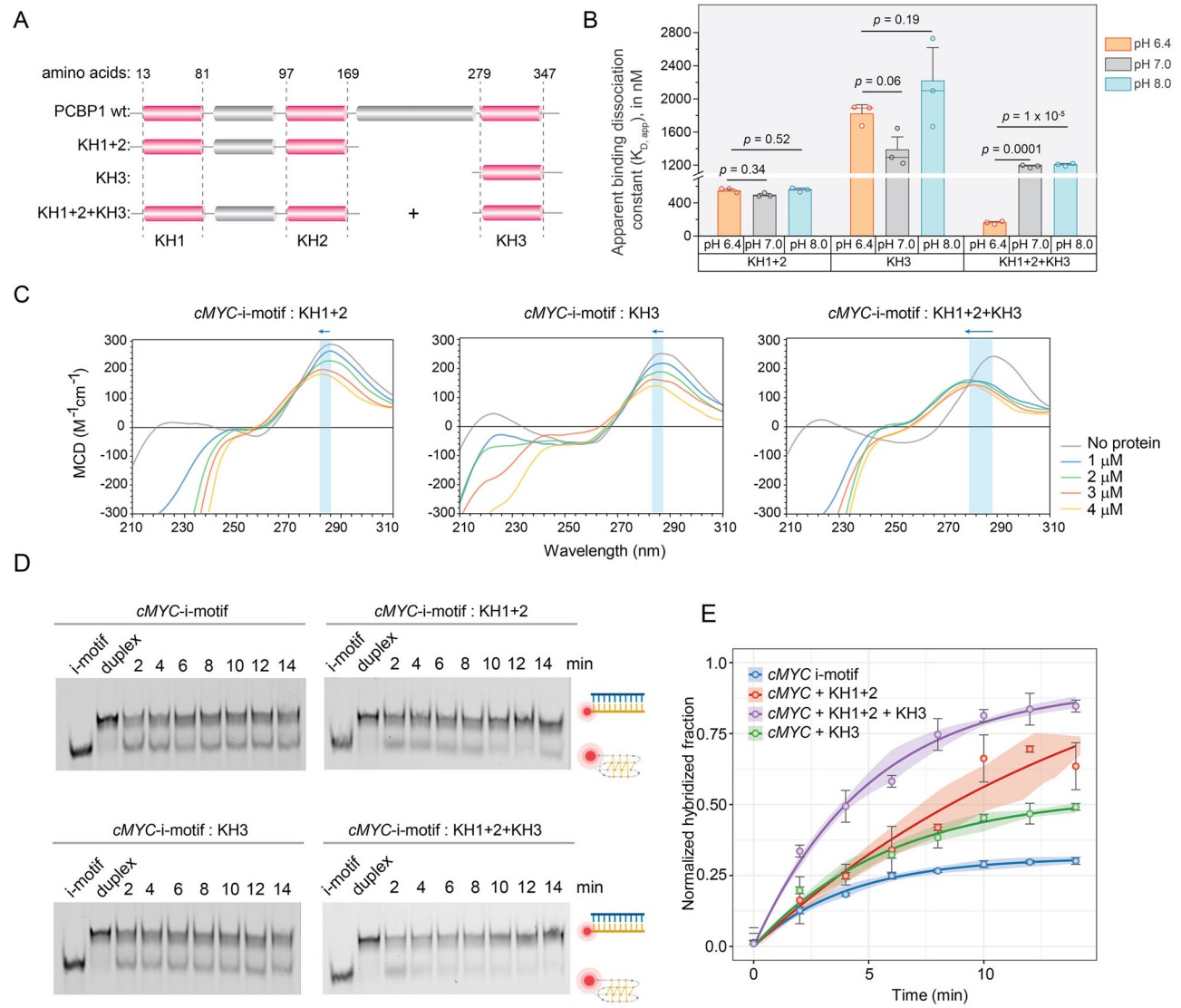

**Fig. 6 | Role of KH domains in i-motif binding and unfolding in vitro.**
**A** Schematic representation of purified wild-type PCBP1 and KH mutants (KH1 + 2 and KH3). KH domains are marked in pink, and the unstructured interconnecting domains are marked in gray. **B** $K_{D,app}$ from MST binding curves between KH mutants and i-motifs at pH 8.0, 7.0, and 6.4; data represent the mean ± SD from three independent biological replicates ($n = 3$). Error bars indicate standard deviation. Statistical significance and $p$ values were calculated using a two-tailed Student's *t*-test, where a $p$ value < 0.05 indicates statistically significant results. **C** CD spectra of *cMYC*-i-motifs upon titrations of KH1 + 2, KH3, and a combination of KH1 + 2 and KH3, with highlighted regions showing blue-shift (with arrows) and

hypochromism. **D** Native PAGE visualizing hybridization reaction aliquots of *cMYC*-iMfs in the presence of different combinations of KH mutants at different time intervals up to 14 min, alongside negative control (i-motifs without complementing strands) and positive controls (hybridized duplex). Uncropped and unprocessed gel images corresponding to native PAGE experiments are provided in the Source data. Created in BioRender. Sabouri, N. (2026) https://BioRender.com/oazozsm. **E** Densitometric analysis of i-motif-trapping. Normalized intensities of hybridized fractions calculated from three independent biological replicates ($n = 3$) at different time points, fitted to single-exponential function equations. Error bars indicate standard deviation.

confirming the feasibility of conducting MST assays between KH mutants and *cMYC*-i-motif at pH 6.4, 7.0, and 8.0 (Supplementary Fig. 24). KH1 + 2 domain bound both i-motifs and their unfolded forms showing lack of structural selectivity, while KH3 showed minimal binding and no preference to either of them (Fig. 6B). KH3 failed to unfold *cMYC*-i-motif, as evident from the negligible blue-shifts in the i-motif CD spectra upon KH3 titrations (Fig. 6C) and the lack of hybridized product accumulation in the i-motif-destabilization trapping assays, similar to the no protein control (Fig. 6D, E). KH1 + 2 also caused a negligible blue-shift in i-motif CD spectra; however, we detected more accumulation of hybridized products starting at 10 min in i-motif-destabilization trapping assays compared to control (Fig. 6C–E), suggesting that KH1 + 2 cannot unfold i-motif as efficiently as PCBP1, but upon interaction, destabilizes i-motif, causing delayed

unfolding. Strikingly, combining KH1 + 2 and KH3 restored full-length PCBP1 function, demonstrating enhanced affinity and selectivity for i-motifs in MST assays (Fig. 6B). This reconstitution also induced significant blue-shifts and hypochromism in CD spectra (Fig. 6C) and increased the accumulation of hybridized products, mirroring the unfolding efficiency like full-length PCBP1 (Fig. 6D, E). These findings highlight that all three KH domains orchestrate selective recognition and efficient unfolding of i-motifs.

## Molecular basis of i-motif unfolding by coordination of KH domains

We proposed two models to elucidate the molecular coordination of KH domains in i-motif regulation. The first model suggested that KH1 + 2 and KH3 form a complex to selectively bind and unfold

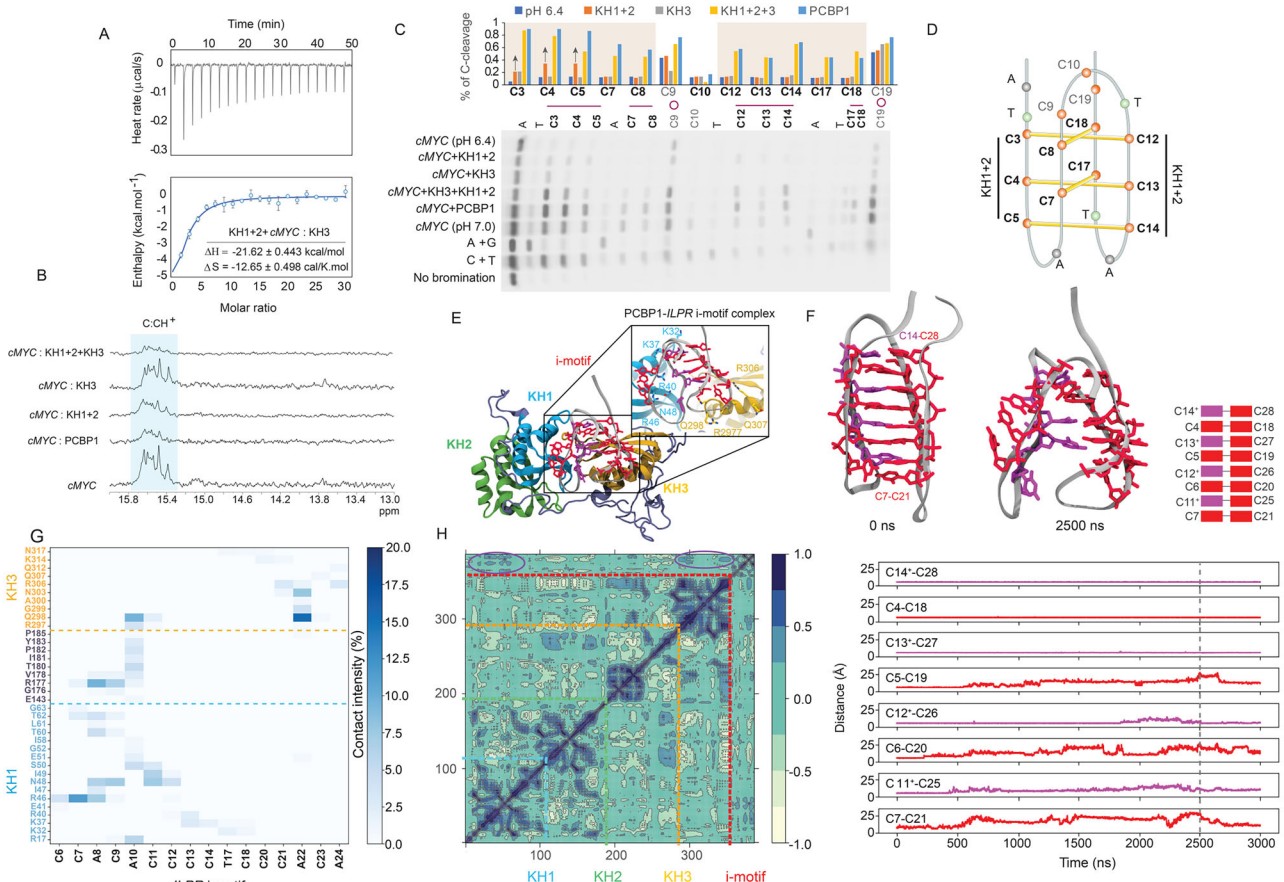

**Fig. 7 | Molecular mechanism of i-motif resolution by coordination of KH domains. A** ITC interaction between a 1:1 mix of KH1 + 2 and *cMYC*-i-motif in the ITC cell, and KH3 in the syringe at pH 6.4. Top: enthalpic heat released versus time at 25 °C during titrations. Bottom: thermogram of integrated peak intensities against molar ratio. Δ*H* (enthalpy) and Δ*S* (entropy) are provided. Data and error bars calculated from mean ± SD from three independent biological replicates (*n* = 3). **B** ¹H NMR spectra showing imino proton resonances (15–16 ppm) of *cMYC*-i-motif during PCBP1 and KH mutants' titrations. **C** Bromine footprinting gel of *cMYC*-i-motif at pH 6.4 and 7.0 in 1:1-bound complexes of PCBP1, KH1 + 2, KH3, and KH1 + 2 + KH3 at pH 6.4. A + G, C + T ladders, and unreacted *cMYC*-i-motif, i-motif sequence are shown; Cs within C:CH⁺ and i-motif-core-connecting loops marked with line and round shapes, respectively. Bar plot shows the % of C-cleavage by piperidine upon bromination in the i-motif. An uncropped and unprocessed gel image corresponding to bromine footprinting experiments is provided in the Source data. **D** Schematic of *cMYC*-i-motif at pH 6.4, based on bromine footprinting; C:CH⁺ base pairs in first C-tract involving C3–C5 and C12–C14 are destabilized

upon KH1 + 2 interactions. **E** In silico modeled complex of PCBP1-bound partially hemi-protonated *ILPR*-i-motif with the following modifications: G-to-A at G2, G16, and G30. C-bases in red and violet are unprotonated and protonated, respectively. KH1-3 domains in PCBP1 are in green, sky blue, and yellow, respectively. Potential amino acids in the interactions within the KH1 and KH3 domains are labeled. **F** During simulation, starting *ILPR*-i-motif structure at 0 ns and i-motif at 2500 ns of simulation are shown. Inter-residual distances between Cs in *ILPR*-i-motif base-pairing throughout simulation (b)−1. **G** Binding mode analysis of PCBP1 with *ILPR*-i-motif throughout 2500 ns simulations with individual contribution of residues within KH1, KH3, and the unstructured region between KH2 and KH3. The darker the blue color of contact intensity (%), the more the frequency of intermolecular interactions. **H** Dynamical Cross-correlation Matrix between the atomic displacements considering Cα for protein residues and phosphorus for i-motif residues during simulation (b)−1. Marked violet circles highlight correlated motions between protein domains and part of the i-motif.

i-motifs. The second model proposed that KH1 + 2 initiates remodeling, facilitating KH3 recruitment for subsequent unfolding. To test the first model, we performed ITC to assess binding thermodynamics. No significant interaction was detected between KH1 + 2 and KH3 in isolation (Supplementary Fig. 25). PCBP1–i-motif interaction was highly exothermic, driven by enthalpic contributions (Supplementary Fig. 26A). KH1 + 2 alone also bound i-motif, displaying a negative Δ*H* (enthalpy) exceeding Δ*S* (entropy). However, full-length PCBP1 exhibited stronger Δ*H*, suggesting additional interactions, potentially involving KH3 (Supplementary Fig. 26A, C). KH3 alone did not bind i-motif, indicating that a prior remodeling by KH1 + 2 was required (Supplementary Fig. 26B). When KH1 + 2 was pre-incubated with *cMYC*-i-motif before KH3 titration, Δ*H* closely matched that of full-length PCBP1, supporting a sequential remodeling and recruitment mechanism, ruling out the first model (Fig. 7A). To validate further, we performed ¹H NMR titrations of KH mutants upon *cMYC*-i-motif,

individually, in combination, and with full-length PCBP1. Consistent with ITC, KH1 + 2 induced significant proton resonance shifts and broadening, indicative of structural changes in *cMYC*-i-motif upon binding. Contrarily, KH3 induced minimal spectral changes, reinforcing its lack of direct binding. However, a combination of KH1 + 2 and KH3 generated pronounced spectral broadening comparable to full-length PCBP1 (Fig. 7B), suggesting cooperative i-motif disruption by KH domains. Bromine footprinting further mapped destabilized C:CH⁺ pairs during protein interactions. KH1 + 2 increased cleavage at C3–C5 and C12–C14 within the first and third C-tracts, while KH3 alone had no significant effect. Their combination enhanced cleavage across all Cs, mirroring the unfolding pattern observed with full-length PCBP1 and the unfolded form at pH 7.0 (Fig. 7C). These suggest that KH1 + 2 destabilizes specific C:CH⁺ pairs in the first and third C-tracts (Fig. 7D), facilitating KH3 recruitment, which ultimately unfolds i-motif.

To illustrate the molecular basis of PCBP1–i-motif interactions, we modeled five PCBP1–i-motif complexes using *ILPR*-i-motif crystal structure[40] as a template (Fig. 7E, Supplementary Fig. 27A) because of its demonstrated ability to bind PCBP1 with moderate unfolding (Fig. 2D, E). However, we reverted three G-to-A mutations (G2, G16, G30) to mimic wild-type *ILPR*-i-motif and performed microsecond-scale MD simulations. To assess the impact of i-motif protonation on PCBP1-mediated unfolding, we modeled each complex under three conditions: (a) fully unprotonated Cs, (b) four protonated C:CH⁺ pairs (C11-C14), and (c) eight protonated C:CH⁺ pairs (C11-C14 and C18-C21). Condition (b) resembled our in vitro experiments at pH 6.4, less acidic and protonated than the published i-motif crystal structure (~pH 5.5)[40]. The simulations revealed that PCBP1 interacts with i-motif in distinct binding modes depending on its protonation. Across all simulations, the DNA-protein complexes maintained relatively stable conformations, as evidenced by the root mean square deviation (RMSD) profiles of PCBP1 and i-motif (Supplementary Fig. 27B, C). C:CH⁺ pairs exhibited greater stability and lower flexibility than C:C pairs, as reflected in the stable RMSD values (~3 Å) observed for (c) (Supplementary Fig. 27D). In condition (a), spontaneous i-motif destabilization was observed in structures 2 and 3 (Supplementary Fig. 28A–D, Supplementary Movies 1 and 2). In (b), the i-motif and its C:CH⁺ pairs remained largely intact, having minor rearrangements. However, in structure 1, an unfolding event was detected at C7–C21 pair propagating in a zipper-like manner (Fig. 7F), as characterized by progressive inter-base distance increases, with C:C pairs unfolding earlier than C:CH⁺ pairs (C6–C20 at ~250 ns, C11–C25 at ~400 ns, C5–C19 at ~500 ns, and C12–C26 at ~1800 ns) (Fig. 7F, Supplementary Movie 3). In contrast, no unfolding was observed in condition (c), reinforcing that higher protonation stabilizes i-motif, delaying PCBP1-mediated unfolding (Supplementary Movie 4). We further examined KH-domain contributions to PCBP1-*ILPR*-i-motif interactions. I-motif primarily contacted KH1 and KH3 via A8–C9–A10 and A22–C23–A24 loops, respectively (Fig. 7G), which acted as initiation points for unfolding. These loop dynamics correlated with key residues from KH1 and KH3 (Fig. 7H). We mapped frequently observed amino acids to KH1, KH3, and the unstructured region between KH2 and KH3, implicating these residues as being involved in i-motif unfolding. Consistent with in vitro findings, these simulations highlight the cooperative role of KH1 and KH3 in i-motif remodeling and unfolding, while underscoring the influence of protonation in regulating i-motif unfolding.

## Discussion

Intracellular regulation of i-motifs is controversial due to their transient nature and sensitivity to non-physiological pH, which complicates efforts to define their protein-mediated interactions and dynamics. Although PCBP-family proteins showed promising in vitro results[21,22,24,26,63], their ability to bind i-motifs has been questioned, likely because they have restricted access to hydrogen-bonding within the i-motif core[20,24,62]. To address this, we systematically examined PCBP1 interactions across diverse i-motifs with varying stability and C-content across different pH, which revealed PCBP1's selective binding to i-motifs, with a distinct preference for highly C-rich sequences. Longer C-tracts strengthened PCBP1's affinity for unfolded forms, compared to folded i-motifs with shorter C-tracts. We also demonstrate that PCBP1 retains appreciable affinity for unstructured C-rich DNA, indicating a degree of binding plasticity that may be relevant under physiological conditions where i-motif stability fluctuates. Nonetheless, quantitative and competitive analyses demonstrate a consistent energetic and competitive advantage for folded i-motifs, suggesting that i-motif stability and C-density together modulate PCBP1 binding. Our findings expand previous studies that focused on individual iMfs[20,22,23], by demonstrating how sequence-context, structure, and protonation collectively influence PCBP1's selectivity. To place our findings in a broader biological framework, we compared

PCBP1 with 19 hnRNP proteins, recently identified as putative i-motif interactors in a proteomics screen[19]. Many hnRNPs share roles in mRNA binding, recognition of poly-CT or poly-AG elements, underscoring the functional diversity within this family (Supplementary Fig. 29). Our findings reveal that PCBP1 unfolds different i-motifs with varied kinetics, consistent with previous reports of its co-localization with i-motif foci in cells[26]—an unlikely observation if PCBP1 indiscriminately resolved all i-motifs. This challenges the view that a single protein universally unfolds all i-motifs, and instead supports a selective regulatory mechanism.

In comparison with other hnRNP proteins, such as hnRNP-K or hnRNP-LL, which both preferentially interact with C-rich sequences, PCBP1 shows a stronger preference for folded i-motifs. Specifically, hnRNP-K can both stabilize[64] and unfold i-motifs, suggesting its sequence-specific functions, whereas hnRNP-LL unfolds i-motif structures[22] although its mechanistic determinants remain poorly defined. By contrast, PCBP1 recognizes residues in the lateral loops of folded i-motifs, promoting their destabilization and unfolding (Fig. 7E, F). PCBP1 remains stable and active across a broad pH range, including a mildly acidic environment that favors i-motif formation, supporting its robust adaptability. Further, hnRNP-K and -LL have mainly been studied in relation to *cMYC* and *BCL2* i-motifs, respectively; therefore, their broader sequence preferences are so far limited.

We also demonstrated that PCBP1-mediated i-motif unfolding is modulated by i-motif protonation status and hairpin-forming propensity (Fig. 8), an aspect not well studied for hnRNP-K or -LL. These differences likely arise from distinct domain architectures. Unlike hnRNP-LL, which utilizes RRM (RNA recognition motif) domains to interact with C-rich sequences[65], PCBP1 and hnRNP-K rely exclusively on KH domains for nucleic acid binding[66,67] (Supplementary Fig. 30A). Structural alignment showed that both proteins share conserved KH folds, yet differ in amino acid composition and domain organization (Supplementary Fig. 30A, B). Specifically, the KH2-KH3 linker is highly flexible, enabling variable KH3 orientations in PCBP1 and hnRNP-K. This is also supported by MD simulations showing KH3's dynamic positioning. Thus, despite conservation within individual KH folds, differences in their domain flexibility, sequence, and interdomain arrangement may underlie distinct mechanisms of i-motif interaction and unfolding by PCBP1 and hnRNP-K (Supplementary Fig. 30C, D). Unlike hnRNP-K, whose individual KH domains independently bind and unfold *cMYC* i-motif[21], PCBP1 requires coordinated action of all three KH domains: KH1 + 2 initiate docking and local rearrangement, allowing KH3 to complete unfolding. Our MD simulations complemented the in vitro results by supporting the dependence of PCBP1-driven unfolding on i-motif protonation. Analysis of interaction frequencies and persistence generated a residue-level interaction heatmap, identifying candidate amino acid residues for future mutagenesis to define their roles in i-motif selectivity and regulation.

A significant challenge in i-motif research is the discrepancy between in vitro and in-cell observations regarding i-motif formation[24]. Sequencing across different cell lines showed considerable variation in genomic i-motif distribution, possibly due to experimental differences and potential non-specific interactions of iMab[44]. To address this, we established an iMab-ChIP workflow using rigorously selected i-motif-forming and moderately C-rich non-i-motif-forming genomic regions, inspired by prior BG4 and iMab-based ChIP-seq and CUT-and-Tag studies[15,45]. Using this approach, we investigated i-motif dynamics in specific loci following PCBP1-KD in HeLa cells. Only a subset of loci–those that behaved consistently with in vitro properties exhibited enhanced i-motif levels, despite robust PCBP1 occupancy across all regions. However, PCBP1-KD downregulated gene expression of all examined i-motif-containing genes, suggesting that their transcriptional outputs reflect a combination of structural effects and broader PCBP1-dependent regulatory pathways.

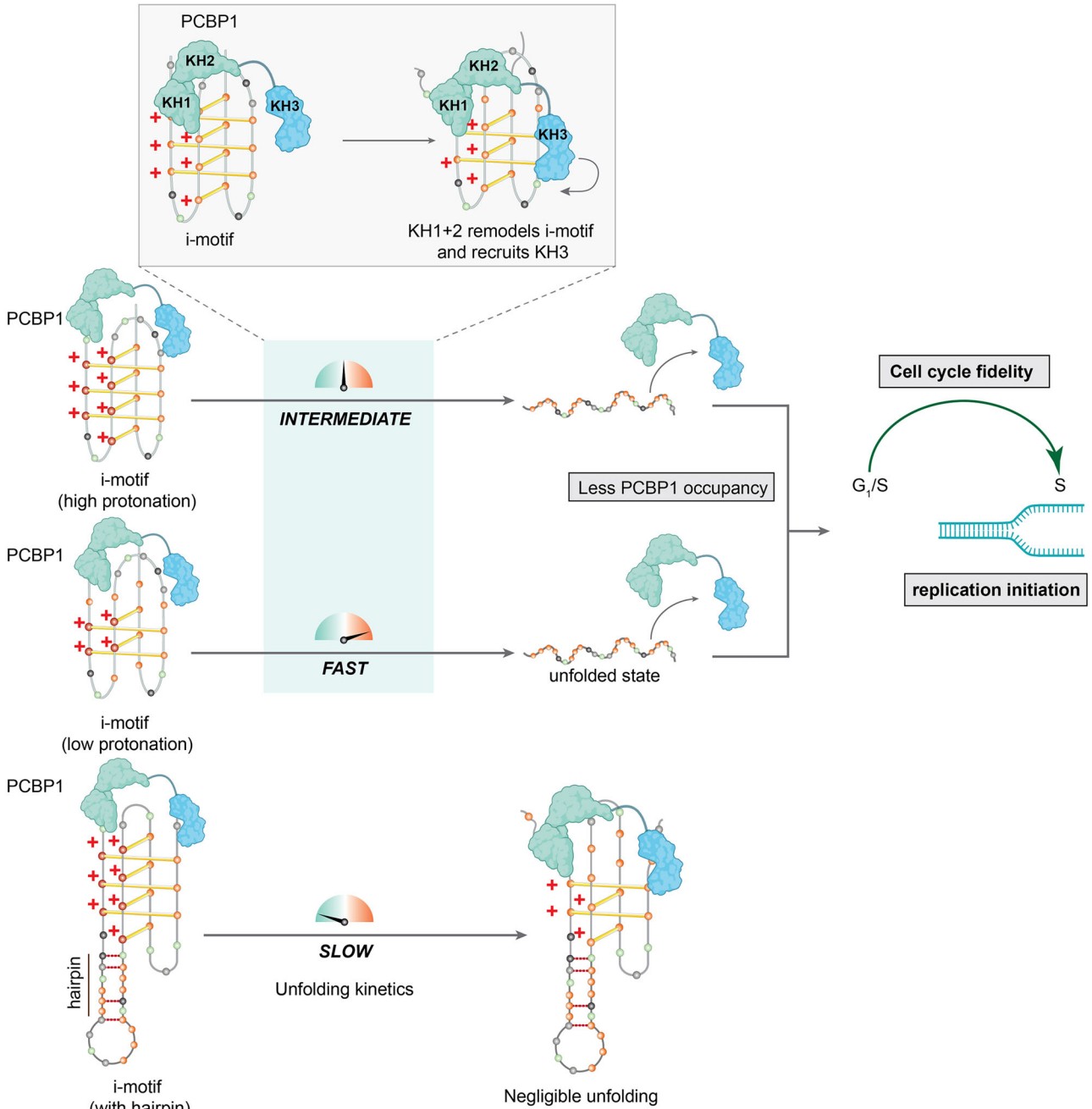

**Fig. 8 | Schematic model of PCBP1-mediated unfolding of selective i-motifs during the cell cycle.** PCBP1 binds i-motif DNA with higher affinity than its corresponding single-stranded forms. Upon binding, PCBP1 promotes i-motif unfolding in a manner that depends on both the protonation state of i-motifs and their propensity to form hairpin conformations. I-motifs that exist in equilibrium with hairpin structures remain accessible to PCBP1; however, stable hairpin formation markedly suppresses PCBP1-mediated unfolding. Protonation further modulates the kinetics of this process, with higher protonation states slowing unfolding, whereas lower protonation accelerates the unfolding. Although PCBP1 can associate with single-stranded forms, these interactions are weaker, and i-motif unfolding consequently reduces PCBP1 occupancy at these genomic sites, thereby permitting replication initiation. PCBP1 contains three KH domains—KH1, KH2, and KH3—that act through a hierarchical two-step mechanism. Initial engagement of the i-motif by KH1–KH2 domains induces local structural destabilization, which subsequently recruits the KH3 domain to drive complete unfolding. This PCBP1-mediated spatiotemporal dynamics of i-motifs is proposed to support genomic integrity and cell-cycle progression by facilitating replication initiation and enabling the $G_1$/S transition into S phase. Created in BioRender. Sabouri, N. (2026) https://BioRender.com/b3vfrkt.

Variability in cellular i-motif detection may also stem from asynchronous cell populations, where cell cycle heterogeneity introduces differences in chromatin state and transcriptional activity. Since i-motif formation is cell cycle–regulated, this heterogeneity may obscure locus-specific dynamics. By synchronizing cells at $G_1$/S and S phases, we captured precise temporal folding patterns in i-motifs while preserving cell cycle phase-dependent changes. PCBP1-regulated i-motifs peaked transiently at $G_1$/S and resolved in early S phase, coinciding with increased PCBP1 occupancy at the same loci at $G_1$/S. These findings highlight the physiological relevance of PCBP1 interactions with both folded and unfolded i-motifs. Because PCBP1 binds folded i-motifs with high affinity, and the PCBP1–i-motif complex remains stable even in the presence of C-rich RNA, transient increases in i-motif abundance at $G_1$/S likely bias PCBP1 toward binding folded structures.

On the contrary, PCBP1–ssDNA complexes are more susceptible to competition with alternative substrates. As i-motifs are generally rare under physiological conditions[24], this narrow temporal window at $G_1/S$ provides an opportunity for PCBP1 to preferentially associate with i-motifs over ssDNA. Thus, PCBP1 acts as a critical regulator of specific i-motifs contributing to the fidelity of $G_1/S$ transition (Fig. 8). PCBP1-KD caused $G_1/S$ arrest and enhanced γH2AX levels, thereby preventing DNA replication. In vitro, PCBP1 facilitated replication on i-motif-forming templates, which otherwise hinder polymerase progression. These observations align with previous reports showing reduced replication efficiency upon failure to resolve i-motif structures and increased γH2AX accumulation upon PCBP1-KD[26]. So far, there is no direct evidence that other hnRNP proteins, such as hnRNP-K regulates i-motif dynamics during $G_1/S$, although it is reported to influence cell cycle progression through distinct mechanisms, such as stabilizing *CDK6* transcripts[68], or transactivating cell cycle arrest genes (*14-3-3σ*, *GADD45*, and *CDKN1A/p21*)[69].

We simultaneously investigated G4 dynamics upon PCBP1-KD using BG4-ChIP at the same loci because i-motif formation influences G4 stability on complementary strands and PCBP1 has been implicated in recruiting G4 helicase-DHX9 to G4-forming regions[27]. We observed significant changes only at *cMYC*, while *BCL2* and *ILPR* remained unaffected. G4 structures at all loci appeared more robust than their i-motif counterparts, aligning with in-cell NMR findings that i-motifs are highly transient in cells[24]. Under synchronized conditions, only *cMYC* showed expected G4 fluctuations—reduced at $G_1/S$ and increased in S phase—whereas *BCL2* and *ILPR* did not change. While our iMab-ChIP partially aligned with expected trends for i-motif dynamics upon PCBP1-KD at *cMYC* and *ILPR*, our BG4-ChIP results revealed reduced G4 levels at *cMYC*, contrary to recent reports suggesting PCBP1-KD enhances G4 abundance by impairing DHX9 recruitment[27]. These results suggest that PCBP1's influence on G4 dynamics may be indirect, potentially mediated by its impacts on i-motifs or broader role in cell cycle regulation. Thus, our findings challenge simple models proposing universal reciprocal regulation between i-motifs and G4s or generalizable dynamics of these structures across loci during the cell cycle. In fact, they highlight the need for caution when interpreting cellular i-motif or G4 dynamics, which may vary by locus, cell type, and physiological context. While our cellular experiments were performed in HeLa cells, which are widely used in mechanistic studies, future validation in other mammalian cell types will be important to determine the extent of conserved versus context-specific regulation. This raises an open question: while G4s and i-motifs may influence each other at shared sites, do they always function in opposition, or can they co-exist or form independently under different conditions?

In conclusion, our study highlights the importance of investigating specific i-motif loci and their spatiotemporal dynamics in relation to protein interactions. PCBP1-mediated i-motif resolution appears critical during $G_1/S$ transition. These insights provide a foundation for further research into PCBP1's roles in maintaining cell cycle fidelity through i-motif unfolding.

## Methods

### Electrophoretic mobility shift assay (EMSA)
PCBP1 binding to folded and unfolded i-motif structures was assessed by EMSA. 5 nM 5′-Cy5-labeled i-motif-forming sequences (iMfs), (Eurofins Genomics Europe; Supplementary Table S1) were incubated with increasing concentrations of purified recombinant PCBP1 in a total volume of 50 μL. Reactions were performed in assay buffers containing either 20 mM Bis-Tris (pH 6.4) (Zellbio B-020-250) or 20 mM Tris-HCl (pH 8.0) (Sigma T1503), supplemented with 100 mM NaCl and 0.25 mg/mL bovine serum albumin (BSA) (Merck A9647). Samples were incubated at 25 °C for 30 min, after which 10 μL of 80% glycerol was added, and 20 μL of each reaction mixture was resolved

on 8% native PAGE (Polyacrylamide gel electrophoresis) in 50 mM Bis-Tris (pH 6.4) or 50 mM Tris-HCl (pH 8.0), supplemented with 100 mM NaCl, matching the respective assay buffer conditions and pH. Electrophoresis was conducted at 70 V at 6 °C, using a 1× running buffer of identical composition. Gels were imaged using an Amersham Typhoon scanner (Cy5 laser; 300 V; 25 μm resolution) and processed in ImageJ. All experiments were performed in three biological replicates.

### Thermal-shift assay
pH tolerance of purified PCBP1 and KH mutants (KH1 + 2 and KH3) was assessed using a Sypro-orange-based protein thermal-shift dye kit (Applied biosystems; 4461146). Protein samples (10 μM; 20 μL) were prepared in 20 mM MES buffer (pH 5.5, 6.0, or 6.4) or 20 mM Tris-HCl buffer (pH 7.0 or 8.0), each supplemented with 100 mM NaCl, and loaded into flat-bottom 96-well plates (Bio-Rad). Thermal unfolding was monitored using a CFX96 real-time PCR system (Bio-Rad) by heating from 25 to 95 °C at 0.2 °C increments while recording SYPRO Orange fluorescence. The melting temperatures ($T_m$) (Supplementary Tables S3 and S5) were determined from the maximum of the negative first derivative of fluorescence with respect to temperature ($-dF/dT$), corresponding to the midpoint of the unfolding transition. Proteins prepared in storage buffer (20 mM Tris-HCl, pH 8.0, 150 mM NaCl) were used as reference controls. All measurements were performed in three biological replicates.

### Microscale thermophoresis (MST)
We performed MST to study the interaction between PCBP1/KH mutants and 5′-Cy5-labeled iMfs and other oligonucleotides (Supplementary Table S1) and calculate the apparent binding affinities ($K_{D,app}$). 40 nM 5′-Cy5-labeled oligos were incubated in 20 mM MES (pH 6.4) or phosphate buffer (pH 7.0) or Tris-Cl (pH 8.0) with 100 mM NaCl and varying concentrations of PCBP1 or KH mutants (35 nM to 2.5 μM) in 20 μL reactions at 25 °C for 15 min. Then, samples were centrifuged at $11,000 \times g$ for 10 min at 25 °C, and loaded into Monolith NT Capillaries (NanoTemper; SKU:MO-K022). Fluorescence intensities and MST were measured in Monolith NT.LabelFree (NanoTemper) instrument at LED excitation 30%. Data were normalized with MO. Control (v1.6.1) and fitted using 4PL equations that yield the $K_{D,app}$, where C is the inflection point or $K_{D,app}$ in nM.

$$y(x) = A + \frac{B - A}{1 + (\frac{x}{C})^D} \qquad (1)$$

Results were expressed as mean ± SD from three biological replicates.

### Circular dichroism (CD)
We conducted CD experiments to observe i-motif-spectral changes upon PCBP1/KH mutants' interactions, spectral analyses of PCBP1 and KH mutants, and pH-dependent changes in PCBP1 using a Jasco J1700-CD spectrophotometer. All CD scans were performed with 4 μM iMfs or proteins within a 320–210/200 nm range and at 100 nm/min scanning speed. Data points were obtained at 0.5 intervals and averaged over three accumulations. We used 1 mm-pathlength quartz cuvettes (Hellma) carrying 200 μL samples. Digital integration time and bandwidth were 2 s and 1 nm, respectively. Oligos were first annealed into 20 mM MES (pH 6.4) and 100 mM NaCl.

To monitor i-motif-spectral changes upon interactions, PCBP1 or KH mutants were titrated into iMfs in increasing concentrations (1–4 μM) and incubated for 30 min at 25 °C.

To measure the transition pH ($pH_T$) of i-motifs, we acquired i-motif spectra (4 μM) within a pH range from 5.4 to 7.2. CD values were corrected for respective buffer contributions and converted to molar ellipticities (MCD) using the following Formula (2), where θ is CD ellipticity in millidegrees, *c* is DNA concentration (mol/L), and *l* is

pathlength (cm):

$$\Delta\varepsilon\left(\text{M}^{-1}\text{cm}^{-1}\right) = \frac{\theta}{32980 \times c \times l} \quad (2)$$

$pH_T$ values were calculated by plotting fraction of folded i-motifs (calculated from MCDs at 287 nm (positive maxima)) vs pH, using the equation below:

$$f(\text{folded}) = \frac{\text{MCD}_i - \text{MCD}_{\text{pH7.2}}}{\text{MCD}_{\text{pH5.4}} - \text{MCD}_{\text{pH7.2}}} \quad (3)$$

We fitted these data points using the Slogistic1 Eq. (1) in OriginPro 2020, where $x_c = pH_T$. Results were shown as means ± SD of three biological replicates.

## Concentration-dependent CD melting assay
To assess whether the four i-motifs undergo intermolecular folding at 4 μM concentration under our experimental conditions, we conducted concentration-dependent CD melting assays across a concentration range of 2–24 μM. CD melting experiments were conducted under identical instrumental parameters as described above. Spectra were recorded between 20 and 95 °C at 5 °C intervals with a heating rate of 1 °C/min. To ensure thermal equilibration, each temperature point was maintained for 180 s prior to acquisition. CD signals at the positive ellipticity maximum (-287 nm) were normalized by calculating the fraction of folded using the following relation:

$$\text{Fraction of folded} = \frac{(\text{ellipticity}_i - \text{ellipticity}_{95\,°\text{C}})}{(\text{ellipticity}_{20\,°\text{C}} - \text{ellipticity}_{95\,°\text{C}})} \quad (4)$$

Then melting curves were generated by plotting the fraction of folded values versus temperature range; and fitting the data points using 4PL equations that yield the melting temperatures ($T_m$), $C$ is the inflection point or $T_m$ in °C

$$y(x) = A + \frac{B - A}{1 + (\frac{x}{C})^D} \quad (5)$$

## I-motif-destabilization trapping assay
To estimate the i-motif unfolding kinetics by PCBP1 and KH mutants, we optimized an i-motif- destabilization trapping assay. In this assay, 5′-Cy5-labeled iMfs substrates were prepared at 5 nM in 20 mM MES (pH 6.4), 50 mM NaCl, 2 mM MgCl₂, and 0.25 mg/mL BSA, followed by incubation for 30 min at 25 °C in the presence of 5 nM PCBP1 or KH mutants. Reactions were performed by adding 10 nM i-motif-trap-complementary oligos, specific for each iMfs, at 25 °C upto 20 min. At indicated time points, 10 μL of reaction mix was quenched with 1:1 stop buffer (40% glycerol, 60 mM EDTA, 0.6% SDS, 0.5 μM unlabeled iMfs) and analyzed by 20% native PAGE at 110 V. Gels were imaged using an Amersham Typhoon scanner with a Cy5 laser at 300 V with a resolution of 25 μm per pixel and quantified with ImageJ. Negative controls lacked i-motif-trap-complementary oligos, while in positive controls, iMfs were incubated with i-motif-trap-complementary oligos for 24 h at room temperature to allow complete hybridization. The fraction of hybridization between unfolded i-motif and its complementary trap oligos was estimated and normalized by dividing the intensity of trapped or hybridized products by the sum of trapped and unhybridized band intensity. Data points plotted against time points were fitted using a single-exponential function Eq. (4), to calculate the rate of i-motif-unfolding ($k$), where $F_{max}(t)$ denotes the fractional maximal value (or normalized signal) at time $t$. Experiments were repeated three times.

$$F_{max}(t) = 1 - e^{-kt} \quad (6)$$

## Bromine footprinting
To map the destabilized C:CH⁺ pairs in cMYC-i-motif upon PCBP1 or KH mutant binding, we performed a bromine footprinting assay. 10 μM of 3′-Cy5-labeled cMYC-iMfs were annealed in 20 mM MES (pH 6.4) or 20 mM phosphate buffer (pH 7.0) with 50 mM NaCl by heating at 95 °C for 5 min, followed by slow-cooling overnight. 5 μM iMfs pre-incubated with or without 5 μM PCBP1/KH mutants for 30 min at 25 °C were treated with molecular bromine formed in situ by mixing equimolar KBr (Sigma Aldrich; 221864) and KHSO₅ (Sigma Aldrich; 228036) (20 μM) for 20 min at room temperature. Reactions were stopped by 80 μL of 0.3 M sodium acetate (pH 7.0) and 25 μg/mL calf thymus DNA on ice. Unreacted bromine was removed by two successive ethanol precipitations. The DNA pellet was resuspended in 100 μL of 100 mM piperidine (Sigma; 104094), heated at 90 °C for 20 min to induce bromination-specific strand cleavage, then washed thrice and dried using speed-vac. Samples were dissolved in 95% formamide (VWR; A2156) and 20 mM EDTA, heated at 95 °C for 5 min, and snap-cooled. Purine- and pyrimidine-specific reactions were performed using 4% formic acid (Sigma Aldrich; 5438040100) and hydrazine (Sigma Aldrich; 309400) to generate A + G and C + T sequencing markers, respectively, following Maxam-Gilbert reactions[70]. Samples were analyzed on 20% denaturing PAGE at 60 W, imaged using Amersham Typhoon scanner with Cy5 laser (500 V, 25 μm pixel resolution), and quantified with ImageJ. Each assay was performed in three biological replicates.

## Isothermal titration calorimetry (ITC)
Thermodynamic binding profiles between iMfs and proteins (PCBP1 and KH mutants) were determined using ITC with a MicroCal Auto iTC200 at 25 °C at both pH 8.0 and pH 6.4 using similar buffer conditions. For the binding studies involving KH1 + 2 and KH3, 5 μM of KH1 + 2 was prepared in 20 mM MES (pH 6.4) and 150 mM NaCl and placed in the calorimetric cell. The syringe contained 350 μM KH3 in the same buffer. To investigate the interactions between KH mutants (KH1 + 2 and KH3) and i-motif, 5 μM KH1 + 2 was pre-incubated with an equimolar concentration of cMYC-i-motifs for 30 min before being placed in the cell, while the syringe was filled with 350 μM KH3. For binding studies between proteins and iMfs, 5 μM of each protein was loaded into the cell, and 300 μM of iMfs was placed in the syringe. Control experiments were conducted simultaneously by injecting the same concentrations of substrates into a buffer without ligand to account for the heat of dilution. Oligonucleotides were injected 20 times at 150-s intervals into the calorimeter cell to achieve binding saturation. Data analysis was performed using the Malvern Microcal PEAQ-ITC software, employing a "one-site" binding model to obtain the best-fit values for the number of binding sites ($N$), changes in enthalpy ($\Delta H$), entropy ($\Delta S$), and free energy ($\Delta G$) of binding reactions.

## Purification of recombinant PCBP1 and KH mutants
cDNAs of PCBP1, KH1 + 2, and KH3 were cloned into the pET-His1a vector for overexpression of recombinant PCBP1 in Escherichia coli BL21(DE3) cells. PCBP1 was induced at 18 °C before harvesting by centrifugation. The cell pellet was resuspended in lysis buffer (50 mM NaP 8.0, 500 mM NaCl, 10% glycerol, 0.2% Triton X100, 10 mM imidazole, 5 mM β-mercaptoethanol (β-me), DNase) followed by sonication. The sample was centrifuged (30 min, 20,000 × g, 4 °C), and the supernatant was incubated (1.5 h, 4 °C) with ThermoS NiNTA previously equilibrated in lysis buffer. The mixture was then poured into a gravity flow column and the resin was washed with wash buffer (wash buffer 1: 50 mM NaP pH 8.0, 500 mM NaCl, 10% glycerol, 0.2% Triton

X100, 10 mM Imidazole, 5 mM β-me; wash buffer 2: 50 mM NaP pH 8.0, 1 M NaCl, 10% glycerol, 0.2% Triton X100, 10 mM Imidazole, 5 mM β-me). The elution was performed stepwise with two elution buffers (Elution buffer 1: 50 mM NaP, pH 8.0, 500 mM NaCl, 20 mM imidazole, 5 mM β-me; elution buffer 2: 50 mM NaP, pH 8.0, 300 mM NaCl, 300 mM imidazole, 5 mM β-me). The eluates were analyzed by SDS-PAGE, pooled together according to their purity, and desalted by dialysis in 20 mM NaP, pH 8.0, 150 mM NaCl, 5 mM β-me. His tag was cleaved by TEV protease, and final dialysis was done in 20 mM Tris-Cl 8.0, 150 mM NaCl, 20% Glycerol, 1 mM DTT (Dithiothreitol).

### High-resolution-primer extension assay

To examine the effect of i-motif/hairpin formation to slow down in vitro replication, we performed primer extension assays. 5′-TET (Tetrachlorofluorescein)-labeled primer (1 μM) was annealed to i-motif-forming templates (1.5 μM) in 75 mM NaCl and 6 mM $MgCl_2$ by heating at 95 °C for 5 min followed by slow-cooling overnight. 40 nM of annealed samples were incubated in 50 μl reactions containing 20 mM MES (pH 6.0), 6 mM $MgCl_2$, 0.2 mg/ml BSA, and 0.05 μU/L Klenow fragment (Thermo Scientific; EP0051) with/without 40 nM PCBP1 or KH mutants for 20 min at 25 °C. Primer extension was started with 0.2 mM dNTPs and continued for specific time points. Reacted samples were collected at designated time intervals, quenched with stop solution (95% formamide, 20 mM EDTA), denatured at 95 °C for 5 min, and snap-cooled. 5 μl was loaded on 12% denaturing PAGE (8 M urea (VWR; 28877.292), 25% formamide, 1×TBE), and run at 60 W in 1× TBE running buffer. The gel was visualized using Amersham Typhoon (GE Healthcare) with Cy3 laser (500 V, 25 μm pixel resolution) and quantified using ImageJ. Each experiment was performed in three biological replicates.

### ¹H 1D NMR

NMR experiments were conducted using a Bruker DRX 850 MHz NMR spectrometer, equipped with a 0.7 mm ultra-fast MAS probe, to understand the impact of PCBP1/KH mutants binding on i-motif's structural dynamics. 100 μM of iMfs were prepared and annealed in 20 mM MES buffer (pH 6.4) containing 100 mM NaCl in a solvent mixture of 90% water and 10% $D_2O$. The 1D ¹H NMR experiments were performed in 3 mm NMR tubes (Bruker; Z172598) with an active sample volume of 200 μL. Spectral referencing was performed using an internal standard, TSP [3-(trimethylsilyl)-2,2′,3,3′-tetra-deuteropropionic acid] (Sigma; 450510), set at 0.0 ppm. In the 1D proton spectra, imino proton resonances corresponding to C:CH⁺ bonds in the i-motifs were observed at 15–16 ppm, while resonances indicative of hairpins involving Watson-Crick base pairs were detected at 12–14 ppm. These measurements were obtained using the standard Bruker pulse program "zgesgp" with a spectral width of 20 ppm, 256 scans (ns), acquisition time 2 s, and a calibrated pulse length (pl) of 9.48 μs. NMR titrations were performed by incrementally adding aliquots of purified proteins (PCBP1 and KH mutants) to the 100 μM iMfs. Samples were thoroughly mixed and allowed to reach thermal equilibrium. Proton spectra were recorded at each titration point, following an incubation period of 15 min at 298 K. Data acquisition and processing were carried out using Topspin 4.2.0 (Bruker).

### Docking and MD simulations

MD simulations were performed using the Amber20 package on the structure obtained from Alphafold[71] and RoseTTAFold[72] or RoseTTAFoldNA[73]. Amberff14SB[74] and parmbsc1[75] force fields were used to model protein and DNA i-motif, respectively, with the CUFIX correction for ionic interaction[76] and Amberff14IDPSFF[77] force field for the intrinsically disordered part of PCBP1 between domains 2 and 3 (residues 170–268). The protonation state of the amino acids of the PCBP1 protein was determined on the basis of PropKa[78] calculations. All the systems were solvated in a cubic box of TIP3P water with a solvent buffer of at least 12 Å and about 0.10 M of NaCl salt, taking into account the neutralization of the box. Periodic conditions were applied in combination with the Particle Mesh Ewald method for electrostatics and a cutoff of 10 Å for intermolecular interactions. All the systems were first minimized for 10,000 steps (5000 steepest descent, 5000 conjugated gradient) and then heated from 0 to 300 K during 30 ps in NVT ensemble with a time step of 1 fs. Equilibration (100 ns) and production (1–3 μs) were run in NPT ensemble at 300 K and 1 bar using Langevin thermostat. The SHAKE algorithm was applied to constrain hydrogen-heavy atom bonds and maintain a time step of 2 fs. Trajectories were analyzed using CPPTRAJ[79]. First, MD simulations were performed on 6 conformations of PCBP1 alone, one from AlphaFold prediction, 5 from RoseTTAFold prediction. We selected the representative structure of the most important cluster from a cluster analysis based on the RMSD of the protein residues and a hierarchical agglomerative approach for each trajectory of 100 ns. These structures were combined with the 8AYG[40] structure for i-motif DNA with the ILPR sequence using RoseTTAFoldNA docking. Five docked conformations (Fig. S21A) were selected based on the localization of the DNA towards the different domains, especially KH1 and KH3, and the DNA-RNA binding part of these domains. Three simulation setups were created with only neutral cytosines in the i-motif: one neutral (set a), one with one positively charged cytosine for half of the cytosine pair (set b), and one with one positively charged cytosine per i-motif cytosine pair (set c). The production runs last 1 μs for sets a structures 1, 4, and 5; 2 μs for set a structures 2 and 3, and for all structures of set c; 3 μs for all structures of set b. To compare the structural features of the KH folds between hnRNP-K and PCBP1, we aligned them using VMD software (https://www.ks.uiuc.edu/Research/vmd/)[80].

### Bioinformatic analyses

De novo motif discovery was performed on the bed file extracted from the ChIP-seq datasets using the MEME-ChIP web interface. We used the plotheatmap tool in Galaxy to visualize ChIP-enrichment values around 2000 bp around the TSS (transcription start site). To compare functional overlap and uniqueness between PCBP1 and other hnRNP family proteins, we used previously published i-motif interactome proteomics data[19] that identifies candidate i-motif–binding proteins. From this dataset, all proteins annotated to the hnRNP family were extracted for gene ontology (GO) enrichment of Biological Process (BP) terms using the PANTHER Classification System, applying Fisher's exact test with FDR correction. Enriched GO terms were visualized using a Sankey diagram generated in RStudio 2024.09.0 + 375 and R 4.4.0.

### Cell culture and PCBP1-KD

HeLa cell line was a generous gift from Dr. Sjoerd Wanrooij's group at Umeå University. The cell line was originally procured from ATCC and propagated in academic laboratories. Cells were cultured at the required density in Dulbecco's modified Eagle's medium (DMEM) (Gibco; 10565018) in the presence of 10% FBS (Fetal bovine serum) (Gibco; A5256701) at 37 °C and 5% $CO_2$ in the incubator. PCBP1 knockdown (KD) was performed using small interfering RNA (siRNA) (PCBP1 Human siRNA Oligo Duplex (Locus ID 5093, Origene; SR303372)) transfection with Lipofectamine 3000 (Thermo Fisher Scientific; L3000008) according to the manufacturer's protocol. Briefly, cells were seeded in a 6-well plate at a density of $10^6$ cells per well and allowed to adhere overnight. The following day, 5 ng of siRNA targeting PCBP1 (siPCBP1) or 5 ng of non-targeting scrambled siRNA (siScramble, negative control) was diluted separately in Opti-MEM reduced serum medium (Thermo Fisher Scientific; 31985070). The diluted siRNA was then combined with the diluted Lipofectamine 3000 reagent and incubated for 10–15 min at room temperature. The siRNA-lipid complexes were added to the cells and incubated at 37 °C

with 5% $CO_2$ for 48 h, and the KD was confirmed by western blot analyses with anti-PCBP1 antibody.

## qRT-PCR

To inspect the transcript levels of *cMYC*, *BCL2*, and *ILPR* genes upon PCBP1-KD, we performed quantitative real-time PCR. HeLa cells grown at a density of $1 \times 10^6$ cells per well were treated with PCBP1-KD. Total RNA is isolated from both untreated and KD cells using Qiagen RNeasy Mini Kit (Qiagen; 74104) as per the manufacturer's instructions. cDNA was prepared using UltraScript 2.0 cDNA synthesis kit (PCR Biosystems; PB30.31-02) by incubating a 20 µL reaction containing 2 µg RNA, cDNA synthesis mix, and Ultrascript 2.0 reverse transcriptase at 50 °C for 30 min, followed by a denaturation at 95 °C. qRT-PCR reactions contain 1x SyGreen mix (PCR Biosystems), 0.5 µM forward and reverse primers, and cDNA. The PCR program was 95 °C for 3 min (1 cycle) followed by a three-step reaction of 95 °C for 10 s (denaturation), 57–60 °C for 25 s (annealing), and 72 °C for 20 s (elongation) (35 cycles) and performed in CFX Real Time System (C1000 Thermal cycler (Bio-Rad)). The housekeeping gene, *GAPDH*, is used as an internal control to normalize the variability in target mRNA expression levels, and *HTR6* transcript levels are shown as a negative control. qRT-PCR primers (Supplementary Table S7) are designed using Primer-BLAST, NCBI, and analyzed in OligoAnalyser 3.1-IDT. All experiments are performed in three biological replicates.

## Western blot

To confirm PCBP1-KD and estimate DNA double-strand breaks upon PCBP1-KD, we performed western blot analyses of PCBP1 and γH2AX (H2AX histones phosphorylated at Ser139), respectively, in HeLa cells upon 48 h of PCBP1-KD. Cells were seeded at a density of $7 \times 10^5$ cells per 60 mm dish. At 48 h post-transfection, total protein was extracted from the cells using freshly prepared RIPA lysis buffer (50 mM Tris (pH 8.0), 150 mM NaCl, 1% NP-40, 0.1% SDS, 0.5% sodium deoxycholate) supplemented with Phenylmethylsulfonyl fluoride and protease inhibitor (complete mini EDTA-free tablets, Roche; 5892791001). 30 µg protein lysates were resolved on mini protean TGX precast SDS-PAGE gels (Bio-Rad) and transferred onto nitrocellulose membranes (Amersham™ Protran® Premium Western blotting membranes). Membranes were blocked with 5% non-fat milk in 1× Tris-buffered saline (TBS) with Tween 20 (Amresco; SKU QBIC20726) for 1 h at 4 °C. Then, membranes were briefly washed with 1× TBST and incubated overnight at 4 °C with rabbit monoclonal anti-PCBP1 primary antibody (dilution 1:1000, Abcam; clone number: EPR11049(B), catalog number: ab168377, IgG isotype; Lot number: 1050664-5), or rabbit phosphor-histone H2AX (Ser139) antibody (dilution 1:1000, Cell signaling technologies; Catalog number 2577S; clonality polyclonal; Isotype: IgG; lot number 14) and mouse monoclonal anti-β-Actin antibody (dilution 1:5000, Abcam; catalog number: ab8224; isotype: IgG1; Lot number: 1051636-14). Then, membranes were incubated with HRP-conjugated Goat anti-mouse/anti-rabbit polyclonal IgG (H+L) (dilution 1:5000, Thermo Fisher Scientific; catalog number: 31460) for 2 h at room temperature. Target proteins were visualized on membranes using Western Supersignal West Pico chemiluminescence substrate (Thermo Fisher), and images were captured using ChemiDoc MP system with Image Lab™ software. After incubation with each antibody, membranes were washed for 15 min at room temperature three times with 1× TBST. β-Actin was used as a loading control in the same blots. All experiments are performed in three biological replicates.

## iMab, PCBP1, and BG4 Chromatin immunoprecipitation (ChIP)

$2 \times 10^6$ HeLa cells were crosslinked in 1% formaldehyde (Thermo Fisher; 28908) for 10 min at room temperature, followed by quenching with 125 mM glycine. After quenching, cells were washed with ice-cold 1× phosphate buffered saline (PBS) and collected by centrifugation at $500 \times g$ for 5 min at 4 °C. Cells were lysed on ice in ChIP lysis buffer (50 mM HEPES, pH 7.4; 140 mM NaCl; 1 mM EDTA; 1% Triton X100; 0.1% Na-deoxycholate) for 10 min at 4 °C. Chromatin was isolated and sheared to 200–600 bp using the Covaris E220 system. After RNase A treatment (10 µg/mL) (Thermo Fisher scientific; EN0531) at 37 °C for 30 min, chromatin was incubated overnight (4 °C) with 2 µg iMab antibody (Absolute Antibody) or 500 ng BG4 antibody (in-house prepared)[81] or rabbit monoclonal anti-PCBP1 primary antibody (1:500, Abcam) in ChIP lysis buffer containing 1% non-fat milk in 200 µL, shaking at $25 \times g$. For iMab and BG4, DYKDDDDK Tag (D6W5B) Rabbit mAb anti-FLAG antibody (Cell Signaling, catalog number: 14793S, Clonality: Monoclonal; IgG isotype; Lot number: 7) (8 µg) bound to 10 µL dynabeads protein-G (Invitrogen; 10003D) was prepped by shaking at $130 \times g$ for 2–4 h at 4 °C, then washed and incubated with 80 µL chromatin-iMab complex at $33 \times g$ for 4 h at 4 °C. Beads were washed four times with 100 µL ChIP lysis buffer and twice with 100 µL wash buffer at 4 °C, then immunoprecipitants were eluted with TE buffer (10 mM Tris-Cl, pH 8.0; 1 mM EDTA) containing 0.5 mg/mL Proteinase K by incubation at 37 °C for 1 h and 65 °C for 2 h. Elutes were purified using the ChIP-DNA Clean and Concentrate kit (ZYMO Research; D5201) following the manufacturer's protocol. qPCR was conducted using primers (Supplementary Table S6) targeting promoters with iMfs or non-i-motif-forming control regions. ChIP enrichment was quantified using the percentage of input method, where input DNA corresponds to a fraction of total chromatin collected before immunoprecipitation. Rabbit anti-FLAG IgG was used as the mock control for iMab and BG4, and anti-rabbit IgG was used for anti-PCBP1 ChIP. For each sample, the percentage of input was first calculated for both target loci and negative control regions. Fold enrichment of target loci was then determined by dividing the percentage of input of the target by the corresponding negative control. This two-step calculation allowed us to account for both IP efficiency and background signal, ensuring that the observed enrichment was specific to i-motif-forming regions. The raw $C_t$ values of input, immunoprecipitated (IP) samples, and mock (IgG) in ChIP experiments are provided in detail in Supplementary Data 1–6 alongside the two-step calculations.

## Flow cytometry-based cell cycle analyses

HeLa cells were grown to a density of $1 \times 10^6$ cells per well. Were subjected to a double-thymidine block or PCBP1-KD, followed by trypsinization and subsequently washed with cold 1× PBS. The cells were then fixed with chilled 70% ethanol and stored at −20 °C overnight. After fixation, the cells were centrifuged at $850 \times g$ for 15 min at 4 °C. The resulting pellets were resuspended in 500 µL of FxCycle PI/RNase staining solution (Invitrogen; F10797) and incubated in the dark at 4 °C with gentle rocking for 4 h. DNA content was analyzed using BD CSampler Plus Software version:1.0.34.1. Gating of single cells, histograms of DNA content (Propidium iodide (PI) fluorescence intensity) were analyzed in FCS Express. The percentage of cells in each phase of the cell cycle ($G_1$, S, and $G_2$/M) was determined by fitting the data into Cox multivariate analyses.

## Cell synchronization using the double-thymidine block

HeLa cells were synchronized at the $G_1$/S boundary using a double-thymidine block. Thymidine (2 mM final concentration) (Sigma Aldrich; 89270) was added to the cells grown in 30–40% confluency into DMEM supplemented with 10% FBS overnight, and incubated for 18 h. Cells were then washed thrice with 1× PBS and released into fresh, thymidine-free medium for 9 h. Subsequently, a second thymidine treatment (2 mM final concentration) was applied for another 15–18 h to achieve synchronization at the $G_1$/S transition. After the second block, cells were either harvested at the $G_1$/S boundary after 2 h or released into fresh medium for 4 h to obtain synchronized populations in S phase. Synchronization efficiency was confirmed by flow cytometry analysis of DNA content as described above.

## Anti-BrdU dot-bot assay

HeLa cells were synchronized at $G_1$/S using the aforementioned double-thymidine block and released into fresh medium at four time points at an interval of 1 h upto 4 h. 1.3 μg genomic DNA isolated from cells at each time-point using DNeasy Blood & Tissue Kit (Qiagen; 69504), and were heated at 95 °C and immediately cooled on ice. 0.3 M final concentration of NaOH was added, and DNA was loaded onto a Hybond-N⁺ membrane (GE Healthcare) using a Bio-Dot Microfiltration Apparatus (Bio-Rad). The membrane was blocked overnight with 1% non-fat milk in 1× PBS and incubated with Rat anti-BrdU BU1/75 (ICR1) antibody (Abcam; catalog number: ab6326, Clonality: monoclonal, subtype: IgG2a) (1:1000 dilution) diluted in 1% non-fat milk. After washing the membrane for 15 min twice with 1× PBS, it was incubated with goat anti-rat IgG antibody peroxidase (Merck; catalog number: DC01L-200UG) (1:5000 dilution) diluted in 1% non-fat milk. Dots were developed using Western Supersignal West pico chemiluminescence substrate (Thermo Fisher; 34580), and images were captured using ChemiDoc MP system with Image Lab™ software. Dot-blot assays were repeated in three biological replicates.

## Statistical analyses

All statistical analyses were performed using OriginPro 2020. One-way ANOVA followed by Tukey–Kramer post hoc test was used for ChIP enrichment analyses to compare multiple groups. In qRT-PCR studies and comparison of $K_{D,app}$ between PCBP1/KH mutants and multiple i-motifs, statistical significance was assessed using a two-tailed Student's $t$-test. Data from biochemical and cellular studies are presented as mean ± standard deviation (SD) from three biological replicates. A $p$ value < 0.05 was considered statistically significant.

## Reporting summary

Further information on research design is available in the Nature Portfolio Reporting Summary linked to this article.

## Data availability

The data supporting the findings of this study are available within the paper and its Supplementary Information. For ChIP-qPCR experiments, raw $C_t$ values for Input, mock (IgG), and IP of all three biological replicates (having three technical replicates each) and related calculations are provided separately in Supplementary Datasets 1–6. MD simulation trajectories are available at Zenodo (https://doi.org/10.5281/zenodo.17737326)[82]. Source data are provided with this paper.

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

## Acknowledgements

The authors acknowledge Biochemical Characterization Umeå (BMCU) for ITC and CD instruments, Swedish NMR Centre, Protein Production Sweden (funded by the Swedish Research Council) for purification of PCBP1, KH mutants, and BG4 at Umeå University, and Biorender for graphics. The authors thank Wenner-Gren Foundations (UPD2020-0097) and Swedish Cancer Society (24 0907 PT 01 H) for P.S.'s work. N.S. lab received support from Swedish Cancer Society (22 2380 Pj 01 H), Swedish Research Council (VR-MH 2021-02468), and Knut and Alice Wallenberg foundations (KAW 2021.0173). The authors acknowledge the PSMN mesocenter in Lyon (CPER/SYSPROD 2015–2022 project No. 2019-AURA-P5B and AXELERA Pôle de compétitivité) and GENCI—IDRIS (project A0150800609) for computational resources.

## Author contributions

P.S. and N.S. conceived the project and designed the experiments. P.S. and I.O. performed the experimental studies. N.G. performed the MD simulations. All authors contributed to the data analyses. P.S and N.S. wrote the manuscript with input from all authors. N.S. supervised the work. All authors read and approved the manuscript.

## Funding

## Competing interests

The authors declare no competing interests.
