## [Transparent Peer Review file · Nature Communications]

Mechanistic Insights into PCBP1-driven Unfolding of Selected i-motif DNA at G1/S checkpoint

Corresponding Author: Professor Nasim Sabouri

Version 0:

Reviewer comments:

Reviewer #1

(Remarks to the Author)

The study provides novel insights into the role of PCBP1 and is in principle suitable for publication in Nature Communications.

Minor comments:

Line 51 Please either show additional data using Mbody or remove reference 11 (or move to discussion). Confusing as this antibody fragment is not actually used in any of the studies discussed below.

Technical comments:

- ChIP Experiments: The y-axis labeling in the ChIP experiments requires revision. Instead of simply indicating "ChIP," the authors should specify the data representation, which, according to the methods section, is the fold enrichment over negative regions. However, there is an inconsistency between the figure caption (1G) and the methods section regarding the application of this normalization. The caption suggests universal application across all ChIP experiments, while the methods imply its use only for iMab and BG4 antibodies. This discrepancy needs to be addressed and clarified. To enhance data transparency and facilitate a comprehensive evaluation of the ChIP experiments, we recommend including the percentage of input for both immunoprecipitated and mock samples in the supplementary material. This additional information would provide insight into the immunoprecipitation efficiency across different experimental conditions. Furthermore, consistent and accurate labeling of the y-axis across all ChIP-related figures is essential. This should reflect the actual measure being reported (e.g., "Fold Enrichment over Negative Regions" or "% Input," as appropriate). These modifications will significantly improve the clarity and reproducibility of the ChIP experimental data presented in this study.

- EMSA experiments reveal a clear difference in binding affinity between pH 8 and pH 6.4 conditions. However, PCBP1 appears to retain some binding capacity to C-rich non-iM DNA. Of particular interest is the observation that at pH 8, the DNA-protein complex fails to migrate into the gel, remaining in the well. This phenomenon could be attributed to various factors, including complex stoichiometry, protein aggregation, or gel composition. The methods section indicates the use of 15% native PAGE for resolving binding samples. This choice of acrylamide percentage is unusual, given that DNA-protein complexes are typically better visualized on gels with lower acrylamide concentrations (5-8%). The authors should provide a rationale for this methodological decision and consider whether optimization of the assay might offer improved resolution of the binding interactions, particularly between PCBP1 and unstructured DNA. Furthermore, the MST analysis revealed a Kd for unstructured DNA that despite being 2-fold higher than that for iM DNA, presents values below 1 μ M. These affinities are potentially significant in a physiological context. Therefore, a more detailed exploration of the binding characteristics could provide valuable insights into the specificity and physiological relevance of PCBP1 interactions with both structured and unstructured DNA. We recommend that the authors address these points to enhance the robustness of their EMSA data and its interpretation within the broader context of PCBP1-DNA interactions.

- iM destabilization experiments: In Figure 3D, the authors demonstrate the unfolding effect of PCBP1 on the c-myc iM. However, it is unclear whether the kinetics of unfolding were explored to the point where the iM band is no longer observable. This endpoint is crucial for accurate normalization and fitting of the data. Notably, among the tested iMs, only the telomeric iM (Figure S9A) reaches this condition, resulting in a relative standard error that is an order of magnitude lower

than observed for other iMs. The discrepancy between the reported experimental duration and the data presented in the figures requires clarification. The methods section states that experiments were conducted for up to 14 minutes, yet Figure S9A shows kinetic monitoring of the telomeric iM for 20 minutes. This inconsistency should be addressed and corrected. To enhance the robustness of their findings, we recommend that the authors to extend the kinetic analysis for all tested iMs to the point of complete unfolding, or provide a rationale for the chosen endpoint if complete unfolding is not achievable. Addressing these points will significantly strengthen the interpretation of the iM destabilization experiments and provide a more comprehensive understanding of PCBP1's unfolding activity across various iM structures.

- Primer extension assay: The authors' choice of an AT-linker in place of the natural sequence occurring in the gene promoter requires justification. Previous studies have demonstrated that AT-rich sequences can influence DNA conformation and impede polymerase progression, similar to non-canonical structures (10.1006/jmbi.1996.0805; doi.org/10.3389/fmicb.2022.915069). This approach may thus introduce confounding factors and should be addressed. Furthermore, the use of MES buffer at pH 6.0 for the reactions appears inconsistent with the authors' previous findings that PCBP1 exhibits optimal activity at pH > 6.5. This apparent discrepancy warrants explanation and discussion of its potential impact on the results. In Figure S14, the inclusion of the non-iM control signal would significantly enhance the interpretation of PCBP1's effect on iM-forming templates. This addition would provide a crucial baseline for comparison. Additionally, the rationale for terminating binding reactions at 10 minutes for non-iM templates, as opposed to 20 minutes for iM templates, is unclear. This methodological difference may affect the comparability of results between iM and non-iM conditions. Addressing these points will enhance the robustness and interpretability of the primer extension assay data, providing a more comprehensive understanding of PCBP1's interaction with iM-forming sequences in the context of DNA replication.

- Cell cycle experiments: the flow cytometry analysis confirming that synchronization in G1-S and S phases was successful should be included

Reviewer #2

(Remarks to the Author)

This manuscript presents a detailed mechanistic study of PCBP1-mediated unfolding of i-motif DNA structures with biological implication on the G1/S cell cycle transition. Combining biophysical assays, cell-based experiments, and molecular dynamics simulations, the authors reveal a domain-specific model of PCBP1 function. The dissection of KH domain contributions and their coordination based on i-motif protonation is a key strength. The study addresses a relevant and technically challenging area in non-canonical DNA structure biology. The findings enhance our understanding of how protein–DNA interactions support genome stability and establish a functional link between i-motif dynamics and cell cycle progression.

The data support the authors' conclusions, and the methods are sound and presented with details allowing reproducibility. However, as all cellular work was performed in HeLa cells, this limitation should be explicitly acknowledged to avoid overgeneralising to other mammalian systems. The referencing is focus on recent literature (2002-2024) and potentially might benefit from considering original/historical literature.

The analysis is appears rigorous, and differences between i-motif sequences are clearly articulated. Broader conclusions about genome-wide or therapeutic relevance should be slightly tempered given the study's focus on selected loci in a single cell line.

Graphical abstract: To improve clarity, you might consider highlighting the protonation aspect more clearly, perhaps by using a distinct colour or slightly increased font size.

Line 7: It may be helpful to briefly define PCBP1 (Poly(C)-binding protein 1) at first mention, particularly for readers less familiar with RNA/DNA-binding proteins.

Line 41: A small clarification—if "repair" refers to DNA repair, specifying this could enhance precision.

Line 105: While ILPR is a well-known i-motif-forming sequence, to the best of my knowledge, it's not classified as an oncogene. Rephrasing it as a promoter sequence in the insulin gene would help ensure accuracy or to summarise with the other sequences as i-motif forming sequence in promotor regions.

129 & Figure 2 legend: Since "telomere" can imply the full chromosomal end structure, it might be clearer here to refer to the sequence as the "C-rich telomeric strand" or "hTeloC", in line with the strand-specific focus of the study.

Typical i-motif nomenclature always uses a lower-case i rather than an I. We suggest this is changed throughout the manuscript.

Reviewer #3

(Remarks to the Author)

Reviewer #4

(Remarks to the Author)

Reviewer Report – Nature Communications Review Paper: Mechanistic Insights into Poly-(rC)-binding protein 1 driven Unfolding of Selected i motif DNA at G1/S checkpoint

While the formation of i-motif DNA structures from cytosine-rich sequences has been well documented in vitro, their existence and function in human cells remain highly controversial. This skepticism primarily arises from the strong pH dependency of i-motif formation, which requires acidic, non-physiological conditions for the protonation (hemi-protonation) of cytosine–cytosine base pairs. Despite this limitation, numerous studies have attempted to demonstrate a role for i-motifs in DNA transactions, often using small molecules or sequence modifications. However, such efforts face criticism due to the lack of selectivity of these molecules and because sequence modifications may also affect the complementary G-quadruplex (G4) structures, further complicating interpretations. Therefore, identifying factors that can selectively modulate i-motif formation within cells remains a significant challenge.

In this article, Sengupta and colleagues use a combination of in vitro, cellular, and molecular modeling approaches to propose that PCBP1, a cytosine-binding protein, plays a role in modulating i-motif structures in human cells—especially during the initiation of the replicative phase. They propose that this activity helps to prevent DNA polymerase stalling caused by i-motif structures, a process potentially contributing to genome instability.

Major Comments:

The manuscript is generally well written and the experimental approaches are of good quality, covering diverse strategies that support the main conclusions. The study is structured into three main experimental blocks:

1. In vitro assays to assess PCBP1 interaction and unfolding activity on i-motif structures,
2. Cellular assays to determine the impact of PCBP1 on i-motif formation, and
3. Molecular modeling to propose a mechanistic model of PCBP1 interaction.

However, several concerns limit the novelty and impact of this work:

1. Novelty of PCBP1's Role: Similar studies have already been conducted with other hnRNP family members, such as hnRNP K and hnRNP LL. Notably, the involvement of KH domains in i-motif unfolding has also been examined. This significantly reduces the novelty of this study. Moreover, the observed differences in how KH domains contribute to i-motif unfolding should have been directly compared to existing data on hnRNP K. A comparative analysis would help assess whether a shared model of action exists for this family of proteins regarding i-motif regulation.
2. Cellular Dynamics and the G1/S Transition: The observation that i-motif abundance may increase during the G1/S phase has been reported by various groups. The current manuscript would have benefited from a deeper exploration of this phenomenon. Is the response specific to PCBP1, or could it be general among i-motif interacting proteins, like hnRNP K? Is there evidence of substantial single-stranded DNA accumulation during G1/S that would favor i-motif formation? Furthermore, why were the transcriptional consequences of PCBP1 depletion not explored?

Other Concerns and Technical Comments:

- The authors re-analyze ChIP datasets to validate PCBP1 binding to cytosine-rich regions with potential to form i-motifs. They also compare PCBP1 and iMab ChIP signals at known i-motif-forming regions. However, ChIP analysis of i-motif structures presents a conceptual issue: most i-motifs are not stable at physiological pH, yet the experiments are conducted at pH 7. Similar results might be observed even at pH 7.5 or 8, raising questions about the specificity of the findings.
- Control regions used for normalization in PCBP1 enrichment studies are described as G/C-rich, but in most cases (except for the c-MYC promoter), they are not comparable to target regions. There is also substantial variability in enrichment levels depending on the control region selected. Raw, unnormalized enrichment data and Ct values from the qPCR assays should be included for transparency—this applies to both PCBP1 and iMab ChIP experiments.
- The reported 4-fold selectivity of PCBP1 for i-motif structures over unstructured sequences could result from electrostatic interactions facilitated by cytosine protonation. Including cytosine-rich sequences that do not form i-motifs as controls could help determine the role of protonation in these interactions. Given the high stability of the C7T3 sequence, its conformational state at different pH levels should also be assessed (e.g., by circular dichroism spectroscopy).
- CD spectroscopy experiments were performed at relatively high oligonucleotide concentrations. Could this have favored the formation of intermolecular i-motifs? This possibility should be discussed or ruled out.

Although the study presents interesting findings, its overall novelty is limited by existing literature. Comparative analyses with related proteins, further investigation into the cell cycle dependence of i-motif dynamics, and additional controls in biophysical and ChIP experiments would greatly strengthen the manuscript.

Version 1:

Reviewer comments:

Reviewer #1

(Remarks to the Author)

The authors have addressed all major comments.

Minor comment: the affinity of PCBP1 for unstructured C-rich DNA. While this affinity is consistently lower compared to iM structures, and this difference has been confirmed through various methods, it remains notably high, lowering the impact of the process. This observation warrants further consideration,

Reviewer #2

(Remarks to the Author)

The authors have addressed all my previous comments thoroughly and have substantially strengthened the manuscript. The resubmission is significantly improved in clarity, methodological rigor, and contextual grounding within the existing i-motif literature. The new experiments, additional controls, and revised analyses convincingly support the conclusions. I find the paper suitable for publication in Nature Communications.

Reviewer #3

(Remarks to the Author)

Reviewer #4

(Remarks to the Author)

The additional experiments carried out by the authors fully address my concerns. The work undertaken to respond to the reviewers' comments is of very high quality, demonstrating a sound scientific approach and strong supporting arguments. Consequently, I support the publication of this article in Nature Communications.

Reviewer #5

(Remarks to the Author)

Reviewer #1 (Remarks to the Author):

The study provides novel insights into the role of PCBP1 and is in principle suitable for publication in Nature Communications.

Minor comments:

Line 51 Please either show additional data using Mbody or remove reference 11 (or move to discussion). Confusing as this antibody fragment is not actually used in any of the studies discussed below.

Response: Thank you. We have removed the mention of iMbody and the corresponding reference in the revised manuscript.

Technical comments:

- ChIP Experiments: The y-axis labeling in the ChIP experiments requires revision. Instead of simply indicating "ChIP," the authors should specify the data representation, which, according to the methods section, is the fold enrichment over negative regions. However, there is an inconsistency between the figure caption (1G) and the methods section regarding the application of this normalization. The caption suggests universal application across all ChIP experiments, while the methods imply its use only for iMab and BG4 antibodies. This discrepancy needs to be addressed and clarified. To enhance data transparency and facilitate a comprehensive evaluation of the ChIP experiments, we recommend including the percentage of input for both immunoprecipitated and mock samples in the supplementary material. This additional information would provide insight into the immunoprecipitation efficiency across different experimental conditions. Furthermore, consistent and accurate labeling of the y-axis across all ChIP-related figures is essential. This should reflect the actual measure being reported (e.g., "Fold Enrichment over Negative Regions" or "% Input," as appropriate). These modifications will significantly improve the clarity and reproducibility of the ChIP experimental data presented in this study.

Response: Thank you for this comment. To ensure consistency across the manuscript, we have updated the Methods section, figures, and figure legends and labelled all y-axes as fold enrichment over the corresponding negative control(s). As additionally suggested by **Reviewer 4** (*see below*), we repeated the ChIP experiments at pH 7.4 instead of pH 7.0. We updated the C_t values for input, immunoprecipitated (IP), and mock (IgG) samples in the folder titled "**Additional data – ChIP raw Ct values.**" Importantly, the main conclusions remain unchanged at this pH. For easier comparison between datasets, we have also added the fold-enrichment ChIP values relative to negative control sites for all samples in the Excel sheets provided in the **Source Data files** for the relevant figures.

- EMSA experiments experiments reveal a clear difference in binding affinity between pH 8 and pH 6.4 conditions. However, PCBP1 appears to retain some binding capacity to C-rich non-iM DNA. Of particular interest is the observation that at pH 8, the DNA-protein complex fails to migrate into the gel, remaining in the well. This phenomenon could be attributed to various factors, including complex stoichiometry, protein aggregation, or gel composition. The methods section indicates the use of 15% native PAGE for resolving binding samples. This choice of acrylamide percentage is unusual, given that DNA-protein complexes are typically better visualized on gels with lower acrylamide concentrations (5-8%). The authors should provide a rationale for this methodological decision and consider whether optimization of the assay might offer improved resolution of the binding interactions, particularly between PCBP1 and unstructured DNA.

Response: We thank the reviewer for this helpful comment. We have followed the suggestion, and repeated the EMSA experiments, and loaded the samples on 8% native PAGE. As shown below in **Fig. R1** and in new **Fig. 2C-D**, optimization of the gels significantly improved the resolution of our EMSA, and confirmed our previous findings. Moreover, no immobile complexes were observed to be clogged in the wells under these optimized gel conditions, ruling out protein aggregation effects.

Fig. R1: EMSA gels at pH 8.0 and pH 6.4 showing 5'-Cy5-labeled i-motif forming sequences (iMfs) with band-shifts upon incubation with increasing concentrations of recombinantly purified PCBP1.

The improved resolution revealed two slow-migrating shifted bands with increasing PCBP1 concentrations at pH 6.4. The second and more slowly migrating bands were most evident with *BCL2*-iMfs. At pH 8.0, where i-motifs are fully unfolded, no multiple slow-migrating bands appeared, suggesting that the multiple bands at pH 6.4 likely arise from DNA structural heterogeneity induced by PCBP1 interactions or transient PCBP1 multimer formation. To clarify the binding stoichiometry, we performed ITC (Isothermal titration calorimetry) (**Fig. R2** and **Supplementary Fig. 5**), which fits into one-site binding model at both pH 6.4 and 8.0, thereby excluding higher-order binding stoichiometries.

Furthermore, the MST analysis revealed a K_d for unstructured DNA that despite being 2-fold higher than that for iM DNA, presents values below 1 μM . These affinities are potentially significant in a physiological context. Therefore, a more detailed exploration of the binding characteristics could provide valuable insights into the specificity and physiological relevance of PCBP1 interactions with both structured and unstructured DNA. We recommend that the authors address these points to enhance the robustness of their EMSA data and its interpretation within the broader context of PCBP1-DNA interactions.

Response: We thank the reviewer for this comment. To address this comment, we performed ITC (Fig. R2 and new Supplementary Fig. 5) and obtained complete thermodynamic profiles of PCBP1–iMfs interactions at both pH 6.4 and pH 8.0. These experiments revealed a free energy difference ($\Delta\Delta G$) of 0.5–0.6 kcal·mol⁻¹ between the two conditions (Fig. R3 and new Supplementary Fig. 9), corresponding to an approximately two-fold higher affinity at pH 6.4 (i-motif folded state), in agreement with the MST data. While this preference is modest, it is potentially biologically meaningful, because such small thermodynamic advantages can be amplified in cells through an increase in the local

Fig. R2: ITC showing intermolecular interactions between PCBP1 and iMfs under pH 6.4 and pH 8.0 at 25 °C. *Top panels:* enthalpic heat released versus time during titrations. *Bottom panels:* thermogram of the integrated peak intensities plotted against the molar ratio of the complex. Best-fit curves using single-site binding models and number of binding sites are provided as N for each binding reaction.

concentration of i-motifs or competition with other substrates. Thus, it may bias PCBP1 occupancy toward folded i-motifs under specific physiological conditions (such as, G₁/S transition when i-motifs are more abundant) while still permitting dynamic exchange.

In fact, our cellular data support this mechanistic link. Both i-motif folding at specific loci and PCBP1 recruitment peak at the G₁/S transition. PCBP1 knockdown (PCBP1-KD) results in persistent i-motif structures and impaired G₁/S progression, indicating that PCBP1 binding resolves these transient structures at a critical cell-cycle checkpoint. Because i-motifs are otherwise rare in cells¹, this temporally restricted folding event provides a window for PCBP1 to preferentially associate with i-motifs than with ssDNA. Our other complementary *in vitro* assays (Circular dichroism (CD) spectroscopy, i-motif-destabilization trapping assays, and primer extension assays) further demonstrate that PCBP1 binding can actively remodel i-motifs into single-stranded forms, facilitating replication *in vitro*. These data combined with elevated iMab signals at target loci and G₁/S arrest upon PCBP1 depletion may be explained by this structural remodeling function.

Fig. R3: ΔG (binding free energy difference) of binding reactions between PCBP1 and iMfs at pH 8.0 and pH 6.0.

To further address this point, we also set up competitive EMSA. Considering that PCBP1 also binds C-rich RNA, we performed this assay using a C-rich RNA substrate against preformed PCBP1–*cMYC*-iMfs complexes at both pH 6.4 and pH 8.0 (**Fig. R4** and new **Fig. 2G**). These experiments demonstrated that folded i-motifs-PCBP1 complexes are more resistant to displacement by C-rich RNA than ssDNA-PCBP1 complexes, suggesting that PCBP1 preferentially interacts with i-motifs even in the presence of its canonical RNA targets. Biologically, this implies that ssDNA-bound PCBP1 is more susceptible to competition by other binding partners of PCBP1 or ssDNA, whereas folded i-motifs provide a privileged binding platform that may underlie the observed cell-cycle specificity.

Fig. R4: Competitive EMSA gels and densitometric analyses at pH 6.4 and 8.0, showing displacement of iMfs-PCBP1 complexes with increasing concentration of C-RNA.

- iM destabilization experiments: In Figure 3D, the authors demonstrate the unfolding effect of PCBP1 on the c-myc iM. However, it is unclear whether the kinetics of unfolding were explored to the point where the iM band is no longer observable. This endpoint is crucial for accurate normalization and fitting of the data. Notably, among the tested iMs, only the telomeric iM (Figure S9A) reaches this condition, resulting in a relative standard error that is an order of magnitude lower than observed for other iMs. The discrepancy between the reported experimental duration and the data presented in the figures requires clarification. The methods section states that experiments were conducted for up to 14 minutes, yet Figure S9A shows kinetic monitoring of the telomeric iM for 20 minutes. This inconsistency should be addressed and corrected. To enhance the robustness of their findings, we recommend that the authors to extend the kinetic analysis for all tested iMs to the point of complete unfolding, or provide a rationale for the chosen endpoint if complete unfolding is not achievable. Addressing these points will significantly strengthen the interpretation of the iM destabilization experiments and provide a more comprehensive understanding of PCBP1's unfolding activity across various iM structures.

Response: We thank the reviewer for this comment. We repeated all destabilization assays with optimized conditions to reliably reach the saturation point, extending time points up to 20 min and using 10 nM i-motif-trap-complementary oligos specific for each i-motif, while other conditions were unchanged. Kinetic data were fitted using a single-exponential function, which provides a more accurate estimate of reaction rates (k). The overall trends and conclusions regarding PCBP1-mediated destabilization remain consistent with the original observations. All responsive i-motifs reached saturation within 20 min except *BCL2*, consistent with our original data showing that PCBP1 does not significantly destabilize *BCL2-wt* i-motif.

- Primer extension assay: The authors' choice of an AT-linker in place of the natural sequence occurring in the gene promoter requires justification. Previous studies have demonstrated that AT-rich sequences can influence DNA conformation and impede polymerase progression, similar to non-canonical structures (10.1006/jmbi.1996.0805; doi.org/10.3389/fmicb.2022.915069). This approach may thus introduce confounding factors and should be addressed.

Response: We thank the reviewer for highlighting the potential structural effects of AT-rich sequences, as discussed by Ruggiero *et al.* 2022² and Lavigne *et al.* 1997³. We chose the AT-linker for two main reasons:

1. Our AT-linker (ATATATATAT) was intentionally kept short (10 nt) to act as a neutral spacer that reduces interactions between adjacent i-motif-forming regions rather than mimicking genomic context. Unlike the extended AAAA/TTTT tracts reported in the above cited studies, this alternating AT pattern has been consistently used in our previous G4/i-motif studies^{4,5} with reproducible outcomes. It provides DNA polymerase with a brief “running start” before encountering the i-motif region, thereby minimizing artefactual pausing and better reflecting physiological replication conditions.

2. Natural linkers are sequence-specific and often contain additional Cs or Gs that may influence folding or polymerase behavior. To address this point more directly, we designed a template with the wild-type *cMYC* linker (*cMYC*-WT) (**Fig. R5A**) and performed primer extension under the same conditions. The C-rich linker caused pausing approximately 4-5 nucleotides upstream of the i-motif region (**Fig. R5B**), illustrating the potential challenge with this approach

Fig. R5: Primer extension assay with *cMYC* template having natural linker. (A) Sequences of TET-primer and TET-primer hybridized with primer extension templates (*cMYC-AT* (AT-linker) and *cMYC-WT* (natural linker)) (B) Primer extension assay with *cMYC* template with natural linker in specific time-points up to 20 min both in presence and absence of PCBP1.

Furthermore, the use of MES buffer at pH 6.0 for the reactions appears inconsistent with the authors' previous findings that PCBP1 exhibits optimal activity at pH > 6.5. This apparent discrepancy warrants explanation and discussion of its potential impact on the results. In Figure S14, the inclusion of the non-iM control signal would significantly enhance the interpretation of PCBP1's effect on iM-forming templates. This addition would provide a crucial baseline for comparison. Additionally, the rationale for terminating binding reactions at 10 minutes for non-iM templates, as opposed to 20 minutes for iM templates, is unclear. This methodological difference may affect the comparability of results between iM and non-iM conditions. Addressing these points will enhance the robustness and interpretability of the primer extension assay data, providing a more comprehensive understanding of PCBP1's interaction with iM-forming sequences in the context of DNA replication.

Response: We appreciate the reviewer's insightful comment regarding the use of MES buffer at pH 6.0 in the primer extension assay. We acknowledge that our previous findings indicated that PCBP1 exhibits optimal activity above pH 6.4. However, the primer extension reaction involves several additional components, such as the Klenow fragment and MgCl₂. It is previously reported that higher Mg²⁺ ions (in millimolar range) destabilize i-motifs⁶, and in our reaction conditions (4 mM MgCl₂ and 40 nM DNA template), the Mg²⁺ concentration is approximately 10⁵-fold higher than that of the DNA. At such concentrations, i-motif formation becomes weakened, such that at pH 6.4 the structures are insufficiently stable to resist Klenow-mediated replication. Reducing Mg²⁺ levels below 4 mM to favor i-motif folding is not feasible either, as Klenow's catalytic activity is dependent on these ions.

To overcome this technical limitation, we needed to optimize the primer extension assay. We found that at pH 6.0, i-motifs remain folded despite the presence of Mg²⁺, thereby allowing us to monitor how efficiently PCBP1 can facilitate unfolding and accelerate replication through i-motif regions. We recognize that this slightly acidic condition may slightly reduce PCBP1's activity. To address this, we performed CD spectroscopy of PCBP1 across the tested pH range (**Supplementary Fig. 2B**). The spectra revealed only a modest (~6%) decrease in the α -helical content accompanied by a compensatory increase in β -structure at pH 6.0 (**Fig. R6** and **Supplementary Fig. 19**) suggesting only a minor structural rearrangement rather than unfolding of the protein. The thermal-stability assay confirms that

PCBP1 remains well-folded, with a melting temperature near 50 °C at pH 6.0 instead of 55°C at pH 6.4, supporting its stability.

Importantly, the retained functionality of PCBP1 at pH 6.0 is supported by including a non-i-motif-forming control template (revised **Supplementary Fig. 17**) which shows no PCBP1-dependent

Fig. R6. Quantitative analysis of PCBP1 secondary structural features (α -helix, β -sheet, turn and others) at different pH values (pH 8.0, 7.0, 6.4, 6.0, and 5.5), calculated from CD spectra using the BeStSel deconvolution algorithm.

effect, whereas the i-motif-forming templates display clear replication enhancement in the presence of PCBP1. For consistency and to address the reviewer's suggestion, the non-i-motif reactions were extended to 20 minutes in the revised experiments. Collectively, these data demonstrate that despite a minor reduction in helical content at pH 6.0, PCBP1 remains structurally competent and functionally active in mediating i-motif unfolding and facilitating replication through i-motif regions under these assay conditions.

- Cell cycle experiments: the flow cytometry analysis confirming that synchronization in G1-S and S phases was successful should be included.

Response: We thank the reviewer for this comment. We have now included the FACS analysis in the supporting information as **Supplementary Fig. 20** and **Fig. R7**.

Fig. R7: Cell-cycle analysis following double thymidine block and release. HeLa cells were synchronized by double thymidine block and collected at the indicated times after release. Representative FSC-A (Forward scatter-area) vs SSC-A (Side scatter area) plots (left) show the gating strategy used to select the main population. DNA content histograms (right) display the distribution of cells in G₁ (red), S (yellow), and G₂/M (blue) phases at 2 h (A) and 4 h (B) post-release.

Reviewer #2 (Remarks to the Author):

This manuscript presents a detailed mechanistic study of PCBP1-mediated unfolding of i-motif DNA structures with biological implication on the G₁/S cell cycle transition. Combining biophysical assays, cell-based experiments, and molecular dynamics simulations, the authors reveal a domain-specific model of PCBP1 function. The dissection of KH domain contributions and their coordination based on i-motif protonation is a key strength. The study addresses a relevant and technically challenging area in non-canonical DNA structure biology. The findings enhance our understanding of how protein–DNA interactions support genome stability and establish a functional link between i-motif dynamics and cell cycle progression.

The data support the authors' conclusions, and the methods are sound and presented with details allowing reproducibility. However, as all cellular work was performed in HeLa cells, this limitation should be explicitly acknowledged to avoid overgeneralising to other mammalian systems. The referencing is focus on recent literature (2002-2024) and potentially might benefit from considering original/historical literature.

Response: We thank the reviewer for the constructive suggestions. We have revised the manuscript to ensure that all interpretations of the cellular data are contextualized specifically within the framework of HeLa cells. Additionally, we have acknowledged this limitation in the discussion section.

Regarding the referencing, we appreciate the suggestion to broaden the historical context. We have now included citations to several foundational studies on i-motif research and PCBP1 protein (see references 6, 7, 29-31 in the main manuscript and also below). We believe that these citations will help to better situate our findings within the broader historical and conceptual development of the field.

6. Mergny, J.-L., Lacroix, L., Han, X., Leroy, J.-L. & Helene, C. Intramolecular Folding of Pyrimidine Oligodeoxynucleotides into an i-DNA Motif. *J. Am. Chem. Soc.* **117**, 8887–8898 (1995).
7. Gehring, K., Leroy, J.-L. & Guéron, M. A tetrameric DNA structure with protonated cytosine-cytosine base pairs. *Nature* **363**, 561–565 (1993).
29. Schmoldt, A., Benthe, H. F. & Haberland, G. Digitoxin metabolism by rat liver microsomes. *Biochem Pharmacol* **24**, 1639–1641 (1975).
30. Piñol-Roma, S. & Dreyfuss, G. hnRNP proteins: Localization and transport between the nucleus and the cytoplasm. *Trends in Cell Biology* **3**, 151–155 (1993).
31. Ji, X. *et al.* α CP binding to a cytosine-rich subset of polypyrimidine tracts drives a novel pathway of cassette exon splicing in the mammalian transcriptome. *Nucleic Acids Res* **44**, 2283–2297 (2016).

The analysis is appears rigorous, and differences between i-motif sequences are clearly articulated. Broader conclusions about genome-wide or therapeutic relevance should be slightly tempered given the study's focus on selected loci in a single cell line.

Response: We appreciate the reviewer's recognition of the rigor of our analysis. We agree that broader interpretations should be drawn cautiously given the focus on selected loci in a single cell line. We have now carefully revised the language throughout the manuscript and acknowledged the comment in our discussion.

Graphical abstract: To improve clarity, you might consider highlighting the protonation aspect more clearly, perhaps by using a distinct colour or slightly increased font size.

Response: Thank you for this suggestion. We have highlighted this by increasing the font size of “+”.

Line 7: It may be helpful to briefly define PCBP1 (Poly(C)-binding protein 1) at first mention, particularly for readers less familiar with RNA/DNA-binding proteins.

Response: We have added a brief definition of PCBP1 at its first mention in the *Introduction* to assist. The revised sentence now reads:

“Here, we investigated the multifunctional protein PCBP1 (Poly(C)-binding protein 1), which belongs to the heterogeneous nuclear ribonucleoprotein (hnRNP) family and is known to bind C-rich RNA/DNA sequences.”

For improved clarity and readability, we have also introduced the full form of PCBP1 upon its first mention in the *Abstract*.

Line 41: A small clarification—if “repair” refers to DNA repair, specifying this could enhance precision.

Response: Yes, this is correct. We have revised the text to refer to “DNA repair” in the introduction and result sections.

Line 105: While ILPR is a well-known i-motif-forming sequence, to the best of my knowledge, it's not classified as an oncogene. Rephrasing it as a promoter sequence in the insulin gene would help ensure accuracy or to summarise with the other sequences as i-motif forming sequence in promotor regions.

Response: Thank you for this helpful comment. We have revised the relevant text in both the main manuscript and figure legends to refer to *ILPR* as an i-motif-forming sequence located in the promoter region of the insulin gene. We have also adjusted the phrasing to group *ILPR* alongside other well-characterized i-motif-forming sequences found in promoter regions of oncogenes such as *c-MYC* and *BCL2*.

Line 129 & Figure 2 legend: Since “telomere” can imply the full chromosomal end structure, it might be clearer here to refer to the sequence as the “C-rich telomeric strand” or “hTeloC”, in line with the strand-specific focus of the study.

Response: We thank the reviewer for this helpful clarification. We have revised the text to refer to the sequence as either the “C-rich telomeric strand” or “hTeloC” throughout the manuscript.

Typical i-motif nomenclature always uses a lower-case i rather than an I. We suggest this is changed throughout the manuscript.

Response: We thank the reviewer for pointing this out. We have carefully revised the manuscript to ensure consistent use of the lowercase “i” in “i-motif” throughout the manuscript except when the term appears at the beginning of a sentence.

Reviewer #3 (Remarks to the Author):

Reviewer #4 (Remarks to the Author):

Reviewer Report – Nature Communications Review Paper: Mechanistic Insights into Poly-(rC)-binding protein 1 driven Unfolding of Selected i motif DNA at G1/S checkpoint
While the formation of i-motif DNA structures from cytosine-rich sequences has been well documented in vitro, their existence and function in human cells remain highly controversial. This skepticism primarily arises from the strong pH dependency of i-motif formation, which requires acidic, non-physiological conditions for the protonation (hemi-protonation) of cytosine–cytosine base pairs. Despite this limitation, numerous studies have attempted to demonstrate a role for i-motifs in DNA transactions, often using small molecules or sequence modifications. However, such efforts face criticism due to the lack of selectivity of these molecules and because sequence modifications may also affect the complementary G-quadruplex (G4) structures, further complicating interpretations. Therefore, identifying factors that can selectively modulate i-motif formation within cells remains a significant challenge.

In this article, Sengupta and colleagues use a combination of in vitro, cellular, and molecular modeling approaches to propose that PCBP1, a cytosine-binding protein, plays a role in modulating i-motif structures in human cells—especially during the initiation of the replicative phase. They propose that this activity helps to prevent DNA polymerase stalling caused by i-motif structures, a process potentially contributing to genome instability.

Major Comments:

The manuscript is generally well written and the experimental approaches are of good quality, covering diverse strategies that support the main conclusions. The study is structured into three main experimental blocks:

1. In vitro assays to assess PCBP1 interaction and unfolding activity on i-motif structures,

2. Cellular assays to determine the impact of PCBP1 on i-motif formation, and
3. Molecular modeling to propose a mechanistic model of PCBP1 interaction.

However, several concerns limit the novelty and impact of this work:

1. Novelty of PCBP1's Role: Similar studies have already been conducted with other hnRNP family members, such as hnRNP K and hnRNP LL. Notably, the involvement of KH domains in i-motif unfolding has also been examined. This significantly reduces the novelty of this study. Moreover, the observed differences in how KH domains contribute to i-motif unfolding should have been directly compared to existing data on hnRNP K. A comparative analysis would help assess whether a shared model of action exists for this family of proteins regarding i-motif regulation.

Response: We thank the reviewer for raising this important point. Although hnRNP-K and hnRNP-LL have been previously reported to bind and unfold i-motifs, our study on PCBP1 identifies important mechanistic differences. Unlike hnRNP-K, where individual KH domains can independently unfold *cMYC* i-motif⁷, our truncation and MD analyses reveal that PCBP1 requires coordinated action of KH1+2 to remodel i-motif and recruitment of KH3 to complete unfolding. We also show a distinct dependence on protonation- and hairpin-propensity for PCBP1 mediated i-motif unfolding, with functional relevance in the cellular context, particularly in G₁/S transition.

Fig. R8: GO (Gene Ontology) enrichment analyses of biological processes (BP) regulated by HNRNP family proteins that are identified to potentially interact with i-motif in the proteomic study by Ban *et al.* The Sankey plot on the left side represents overlapping functions of different hnRNP proteins in nucleic acid binding. On the right shows the gene enrichment bubble plot. Enrichment significance is represented by $-\log_{10}(\text{Pvalue})$. The higher the enrichment score value is (more red), the more significant the pathway is.

To address the reviewer's comment regarding a comparative analysis, we have performed GO enrichment analysis across hnRNP family proteins (Fig. R8 and Supplementary Fig. 29), domain and sequence alignment analysis (Fig. R9 and Supplementary Fig. 30), structure homology modeling (Fig. R9 and Supplementary Fig. 30) alongside literature study, and added a comparative discussion in the revised manuscript.

*GO enrichment analysis:* A recent proteomic study identified 19 hnRNP family proteins[8] as potential i-motif interactors, yet only a few, such as hnRNP-K, hnRNP-LL, and PCBP1 (hnRNP E1) have been functionally examined. To assess shared versus distinct functions among hnRNP family members, we performed GO enrichment analyses of biological processes (**Fig. R8** and **Supplementary Fig. 29**). The Sankey plot illustrates overlap in nucleic acid-binding functions among different hnRNP proteins, and the bubble plot shows significantly enriched biological processes regulated by hnRNPs, where the X-axis indicates enrichment score, bubble size reflects the number of hnRNPs per GO term, and the bubble color represents $-\log_{10}$ P-values (warmer colors = higher significance). These analyses highlight both shared and unique preferences in nucleic-acid–binding among hnRNPs.

*Domain and sequence alignment analysis:* (**Fig. R9A** and **Supplementary Fig. 30A**). hnRNP-LL contains RRM (RNA recognition motif) domains[9], while PCBP1 and hnRNPK lack RRMs and contain solely KH domains[10,11]. Despite the conserved fold of KH domains, their amino acid composition and

domain arrangement differ substantially, influencing sequence specificity and binding mode (**Fig. R9B** and **Supplementary Fig. 30B**).

**Fig. R9**: Different type of nucleic acid binding domains in PCBP1, hnRNP-LL, and hnRNP-K. **(A)** PCBP1 and hnRNP-K contain three KH domains, while hnRNP-K contains an additional KI and proline-rich domain. hnRNP-LL contains four RRM domains. **(B)** Multiple sequence alignment between the KH domains of hnRNP-K and PCBP1 using clustalW. Amino acids highlighted in yellow and marked with * are similar between both proteins. Amino acids highlighted in grey and marked with : are not identical but have similar properties between both proteins. Amino acids highlighted in blue and marked with . are non-identical and share less properties between both proteins. Amino acids that are not marked or blank are different between both proteins. **(C)** Overlay of alphafold-predicted structural models of PCBP1 and hnRNP-K with their respective KH domains and disordered regions marked. **(D)** Alignment on KH1 shows overlay of KH1 and KH2 with highly conserved folds; separate alignment on KH3 likewise shows a well-conserved KH3 domain. The disordered linkers between KH domains differ markedly between the two proteins.

Structure homology modeling analysis: Alignment of AlphaFold-predicted KH domains from PCBP1 and hnRNP-K shows high conservation for KH1 and KH2 (RMSD ≈ 1.3 Å) and for KH3 (RMSD ≈ 0.6 Å). The KH2–KH3 linker, however, is flexible with low alphaFold confidence, leading to variable KH3 orientations (**Fig. R9C-D** and **Supplementary Fig. 30C-D**). MD simulations further support that the relative KH3 position is dynamic rather than fixed. Therefore, while individual KH folds are conserved between PCBP1 and hnRNP-K, differences in their domain flexibility, sequence and interdomain arrangement may underlie their distinct mechanisms of i-motif interaction and unfolding.

All these new analyses have been included in the revised manuscript (**Supplementary Fig. 29, 30**) and in the discussion.

Our data suggest that while hnRNP-K, hnRNP-LL, and PCBP1 share some structural and sequence-binding similarities (*e.g.*, binding to C-rich sequences or resolving i-motifs), they also exhibit key mechanistic differences. PCBP1 binds preferentially to specific residues in the lateral loops of i-motifs to promote unfolding and exhibits a stronger preference for folded i-motif structures. By contrast, hnRNP-K shows both stabilizing¹² and unfolding impact on i-motifs, suggesting its sequence specific functions, whereas hnRNP-LL unfolds i-motif structures¹³ but with poorly defined mechanistic determinants. Our data delineates that PCBP1 unfolding is modulated by i-motif protonation status and hairpin-forming propensity, and that this behaviour is translated in the cellular context playing an impact in G₁/S transition and replication. This aspect is not well studied for hnRNP-K or -LL. Further, we have investigated PCBP1's preference for i-motifs using various i-motifs of different protonation status and stability, while hnRNP-K and -LL have been investigated mainly for *cMYC* and *BCL2* i-motifs respectively, limiting our understanding if they have any preferences for specific i-motif in the genome.

2. Cellular Dynamics and the G₁/S Transition: The observation that i-motif abundance may increase during the G₁/S phase has been reported by various groups. The current manuscript would have benefited from a deeper exploration of this phenomenon. Is the response specific to PCBP1, or could it be general among i-motif interacting proteins, like hnRNP K?

Response: We believe that this phenomenon is very interesting. Our study specifically demonstrates that PCBP1 binds i-motif structures that emerge during the G₁/S transition and selectively unfolds a subset of promoter i-motifs, including those in *cMYC* and *ILPR*. This targeted unfolding promotes expression of S-phase entry genes, and PCBP1-KD results in the persistence of their respective i-motifs, reduced gene transcription, and G₁/S arrest. These results establish a direct mechanistic link between PCBP1-mediated i-motif resolution and G₁/S transition control.

Regarding whether this response may be general to other i-motif–interacting proteins, available literature suggests that hnRNP-K does influence cell cycle progression, but through distinct mechanisms. For example, hnRNP-K stabilizes *CDK6* transcripts, thereby modulating the G₁/S transition¹⁴. It also cooperates with p53 to transactivate cell cycle arrest genes, such as *14-3-3σ*, *GADD45*, and *CDKN1A (p21)*¹⁵. However, there is currently no direct evidence that hnRNP-K regulates i-motif dynamics during G₁/S, and such a comparison remains an open question for future studies. We have now highlighted the available studies on hnRNP-K in the discussion by writing: “*So far, there is no direct evidence that other hnRNP proteins, such as hnRNP-K regulates i-motif dynamics during G₁/S, although it is reported to influence cell cycle progression through distinct mechanisms, such as stabilizing CDK6 transcripts, or transactivating cell cycle arrest genes (14-3-3σ, GADD45, and CDKN1A/p21).*”

Is there evidence of substantial single-stranded DNA accumulation during G₁/S that would favor i-motif formation?

Response: To best of our knowledge, there is no evidence for widespread accumulation of ssDNA during normal, unperturbed G₁/S transition. Instead, our working model is that local regions of the genome may undergo transient strand separation, *i.e.*, through the topological strain generated by replication initiation (*cMYC* contains autonomous replication sites and i-motif-forming sites upstream P1 promoter^{16,17}), negative supercoiling due to transcriptional activity^{18,19} or origin firing²⁰, which creates sufficient opportunity for the C-rich strand to fold into an i-motif.

Furthermore, why were the transcriptional consequences of PCBP1 depletion not explored?

Response: We thank the reviewer for this comment. While our original manuscript focused on the mechanistic aspects of PCBP1–i-motif interactions, we have now assessed the transcriptional effects of

Fig. R10: qRT-PCR studies. Relative expression ($2^{-\Delta\Delta Ct}$) in i-motif-containing and non-i-motif-containing gene expression upon PCBP1-KD.

PCBP1 depletion by qRT-PCR analysis of *cMYC*, *BCL2*, and *ILPR*. All three genes showed significantly decreased transcript levels in PCBP1-KD cells (Fig. R10 and new Fig. 5H). To increase specificity, we further examined the P1 transcripts of *cMYC* and *BCL2*, as this promoter region is known to be directly regulated by G4/i-motif structures^{21,22}. These results indicate that PCBP1 contributes to maintaining the expression of these genes; however, we note that the observed transcriptional effects may arise from both direct and indirect mechanisms, given PCBP1's multifunctional nature and the involvement of other regulators (*e.g.*, Nucleolin, NM23H2, hnRNP-K, hnRNP-LL)^{7,13,23,24} at these promoters. Therefore, while consistent with a possible role in i-motif regulation, these findings should be interpreted cautiously and motivate future studies to delineate direct from secondary effects.

Other Concerns and Technical Comments:

- The authors re-analyze ChIP datasets to validate PCBP1 binding to cytosine-rich regions with potential to form i-motifs. They also compare PCBP1 and iMab ChIP signals at known i-motif-forming regions. However, ChIP analysis of i-motif structures presents a conceptual issue: most i-motifs are not stable at physiological pH, yet the experiments are conducted at pH 7. Similar results might be observed even at pH 7.5 or 8, raising questions about the specificity of the findings.

Response: We have repeated the ChIP experiments using buffers of pH 7.4, which defines physiological pH. The results are consistent with our previous findings. The revised manuscript is now updated with the new ChIP experiments.

- Control regions used for normalization in PCBP1 enrichment studies are described as G/C-rich, but in most cases (except for the c-MYC promoter), they are not comparable to target regions.

Response: We thank the reviewer for this helpful comment. We have revised the sentence to clarify that we selected four negative control regions with moderate C-richness (~50%). It falls within the range of C-content observed in some i-motif loci (for example, *cMYC*), but is previously shown not to form i-motifs. Our rationale for selecting these regions as negative controls was based not only on sequence composition but also on functional evidence. Specifically, these loci were reported by independent i-motif and G4 mapping studies (including ChIP-seq²⁵ and CUT&Tag²⁶ experiments) to show no detectable enrichment for i-motif or G4 structures. As our study represents the first systematic ChIP–qPCR validation using the iMab antibody with multiple controls, we prioritized regions with well-characterized “non-structured” behavior as negative controls.

There is also substantial variability in enrichment levels depending on the control region selected. Raw, unnormalized enrichment data and Ct values from the qPCR assays should be included for transparency—this applies to both PCBP1 and iMab ChIP experiments.

Response: Thank you. We have included the raw, unnormalized enrichment data and C_t values of all ChIP studies conducted in the folder titled ‘**additional data-ChIP-raw Ct values**’.

- The reported 4-fold selectivity of PCBP1 for i-motif structures over unstructured sequences could result from electrostatic interactions facilitated by cytosine protonation. Including cytosine-rich sequences that do not form i-motifs as controls could help determine the role of protonation in these interactions.

Response: We thank the reviewer for this suggestion. To directly test whether the observed PCBP1 binding could result from nonspecific electrostatic interactions with protonated Cs, we included a C-rich control sequence (*mutC*: 5'-TACTACTCACTCACTCACTCACTCAA-3'), previously characterized by Boissieras *et al.*²⁷. This sequence contains 12 Cs but lacks the propensity to fold into an i-motif at acidic pH. CD spectroscopy confirmed the absence of i-motif formation under our experimental conditions (10 mM MES, pH 6.4, 100 mM NaCl), showing a maximum at 274 nm instead of the characteristic 284 nm observed for the *cMYC* i-motif (**Fig. R11A** and **Supplementary Fig. 9A**).

EMSA performed at both pH 6.4 and pH 8.0 using up to 30 nM PCBP1 showed no detectable protein–DNA complex formation with this control sequence, whereas a clear shift was observed for the *cMYC* i-motif as a positive control. These results indicate that PCBP1 binding is not driven by non-specific electrostatic attraction to protonated Cs, but instead depends on the folded i-motif structure. This experiment therefore strengthens our conclusion that PCBP1 selectively recognizes structured i-motifs rather than unstructured C-rich regions (**Fig. R11B** and **Supplementary Fig 9B**).

Fig. R11: I-motif-forming potential and interaction with PCBP1 by *mutC* sequence. (A) CD spectra of *cMYC* and *mutC* in 10 mM MES buffer (pH 6.4) and 100 mM NaCl at 25 °C. Red and grey bars represent their respective positive maxima. (B) EMSA gels at pH 8.0 and pH 6.4 showing interaction between 5 nM of *mutC* and PCBP1 at increasing concentrations alongside positive control of *cMYC* (5 nM) and *cMYC*-bound to 7.5 nM PCBP1 at pH 8.0 and pH 6.4 respectively.

Given the high stability of the C7T3 sequence, its conformational state at different pH levels should also be assessed (e.g., by circular dichroism spectroscopy).

Response: We have confirmed the conformational states of i-motif-forming C7T3 sequence using CD spectroscopy in three different pH (pH 8.0, 7.0, and 6.4). We reconstituted the oligonucleotide sequence in 10 mM MES (pH 6.4) or 10 mM sodium phosphate (pH 7.0), or 10 mM Tris (pH 8.0) buffer supplemented with 100 mM NaCl, and annealed them by heating at 95 °C for 5 min followed by slow-cooling overnight at room temperature. At pH 6.4 and pH 7.0, the positive maxima has been observed at 286.5 and 285.5 nm respectively, defining the signature i-motif spectra. At pH 8.0, the positive maxima have blue-shifted to 276 nm, suggesting that i-motif structure has been unfolded at this pH condition (Fig. R12 and new Supplementary Fig. 7).

Fig. R12: CD spectra of C7T3 in 10 mM MES (pH 6.4) or 10 mM sodium phosphate (pH 7.0), or 10 mM Tris (pH 8.0) buffer supplemented with 100 mM NaCl.

- CD spectroscopy experiments were performed at relatively high oligonucleotide concentrations. Could this have favored the formation of intermolecular i-motifs? This possibility should be discussed or ruled out.

Response: We thank the reviewer for this comment. Although higher oligonucleotide concentrations can, in principle, promote intermolecular i-motif formation, the 4 μM concentration is widely accepted and considered standard practice for CD spectral acquisition, having been routinely employed to characterize i-motif structures^{28,29}. Moreover, structural studies strongly support the predominance of intramolecular folding even at far higher concentrations for certain i-motifs. For example, the crystal structure of the *ILPR* i-motif variant from the Waller lab remains intramolecular, even at 0.3 mM³⁰. NMR studies on *c-MYC* i-motif at 0.5 mM from the Yang lab also shows intramolecular topology³¹.

In the revised manuscript, we directly examined the molecularity of the four i-motifs under our experimental buffer conditions (20 mM MES buffer (pH 6.4) containing 100 mM NaCl) by using concentration-dependent CD melting analyses. Melting temperatures (T_m) were determined within a range from 2 (below which the signal-to-noise ratio was sub-optimal) to 24 μM . In general, intermolecular i-motifs exhibit higher T_m values than intramolecular i-motifs and a strong positive

Fig. R13: Concentration-dependent CD melting assay. Radar plot showing the changes in melting temperatures (ΔT_m) (in $^{\circ}\text{C}$) for *cMYC*, *BCL2*, *ILPR*, and *hTeloC* i-motifs in 10 mM MES buffer (pH 6.4) and 100 mM NaCl in different oligonucleotide concentrations (4, 8, 16, and 24 μM) relative to 2 μM . Radar y-axis denotes ΔT_m values relative to 2 μM . *C6TC6* used as positive control as intermolecular i-motif and its ΔT_m shown at pH 6.4 and at pH 5.0.

correlation between T_m and strand concentration, as demonstrated in prior studies^{32–35}. Therefore, we hypothesized that any intermolecular assembly would manifest as an increase in T_m with concentration. None of the four i-motifs examined exhibited appreciable T_m increase across this range, indicating that they remain intramolecular under our conditions (Fig. R13 and new Supplementary Fig. 6). As a control, we examined *C6TC6*, a well-established intermolecular i-motif³² that shows $\sim 7^{\circ}\text{C}$ T_m elevation between 2 and 24 μM at pH 5.0. Notably, under our experimental pH 6.4, *C6TC6* also displayed no increase in T_m values (Fig. R13 and new Supplementary Fig. 6), consistent with intramolecular conformation, which further supports that our buffer conditions inherently disfavour intermolecular i-motif formation.

Although the study presents interesting findings, its overall novelty is limited by existing literature. Comparative analyses with related proteins, further investigation into the cell cycle dependence of i-motif dynamics, and additional controls in biophysical and ChIP experiments would greatly strengthen the manuscript.

Response: We believe that our above revisions have significantly strengthened our manuscript and enhanced its novelty. We are grateful for the reviewer's constructive suggestions, which have contributed to the improvement of this manuscript.

References:

- Višková, P. *et al.* In-cell NMR suggests that DNA i-motif levels are strongly depleted in living human cells. *Nat Commun* **15**, 1992 (2024).
- Ruggiero, E. *et al.* Human Virus Genomes Are Enriched in Conserved Adenine/Thymine/Uracil Multiple Tracts That Pause Polymerase Progression. *Front. Microbiol.* **13**, 915069 (2022).

3. Lavigne, M., Roux, P., Buc, H. & Schaeffer, F. DNA curvature controls termination of plus strand DNA synthesis at the centre of HIV-1 genome 1 Edited by J. Karn. *Journal of Molecular Biology* **266**, 507–524 (1997).
4. Jamroskovic, J., Deiana, M. & Sabouri, N. Probing the folding pathways of four-stranded intercalated cytosine-rich motifs at single base-pair resolution. *Biochimie* **199**, 81–91 (2022).
5. Sengupta, P., Jamroskovic, J. & Sabouri, N. A beginner’s handbook to identify and characterize i-motif DNA. in *Methods in Enzymology* vol. 695 45–70 (Elsevier, 2024).
6. Zhang, F., Liu, B., Lopez, A., Wang, S. & Liu, J. Opposite salt-dependent stability of i-motif and duplex reflected in a single DNA hairpin nanomachine. *Nanotechnology* **31**, 195503 (2020).
7. Wu, W.-Q., Zhang, X., Bai, D., Shan, S.-W. & Guo, L.-J. Mechanistic insights into poly(C)-binding protein hnRNP K resolving i-motif DNA secondary structures. *Journal of Biological Chemistry* **298**, 102670 (2022).
8. Ban, Y. *et al.* Profiling of i-motif-binding proteins reveals functional roles of nucleolin in regulation of high-order DNA structures. *Nucleic Acids Research* **52**, 13530–13543 (2024).
9. Roy, B. *et al.* Interaction of Individual Structural Domains of hnRNP LL with the *BCL2* Promoter i-Motif DNA. *J. Am. Chem. Soc.* **138**, 10950–10962 (2016).
10. Du, Z. *et al.* Crystal Structure of the First KH Domain of Human Poly(C)-binding Protein-2 in Complex with a C-rich Strand of Human Telomeric DNA at 1.7 Å. *Journal of Biological Chemistry* **280**, 38823–38830 (2005).
11. Yoga, Y. M. K., Traore, D. A. K., Wilce, J. A. & Wilce, M. C. J. Mutation and crystallization of the first KH domain of human polycytosine-binding protein 1 (PCBP1) in complex with DNA. *Acta Crystallogr F Struct Biol Cryst Commun* **67**, 1257–1261 (2011).
12. Ruggiero, E. *et al.* A dynamic i-motif with a duplex stem-loop in the long terminal repeat promoter of the HIV-1 proviral genome modulates viral transcription. *Nucleic Acids Research* **47**, 11057–11068 (2019).
13. Kang, H.-J., Kendrick, S., Hecht, S. M. & Hurley, L. H. The Transcriptional Complex Between the *BCL2* i-Motif and hnRNP LL Is a Molecular Switch for Control of Gene Expression That Can Be Modulated by Small Molecules. *J. Am. Chem. Soc.* **136**, 4172–4185 (2014).
14. Kawasaki, Y. *et al.* MYU, a Target lncRNA for Wnt/c-Myc Signaling, Mediates Induction of CDK6 to Promote Cell Cycle Progression. *Cell Reports* **16**, 2554–2564 (2016).
15. Enge, M. *et al.* MDM2-Dependent Downregulation of p21 and hnRNP K Provides a Switch between Apoptosis and Growth Arrest Induced by Pharmacologically Activated p53. *Cancer Cell* **15**, 171–183 (2009).
16. Vassilev, L. & Johnson, E. M. An Initiation Zone of Chromosomal DNA Replication Located Upstream of the *c-myc* Gene in Proliferating HeLa Cells. *Molecular and Cellular Biology* **10**, 4899–4904 (1990).
17. Waltz, S. DNA replication initiates non-randomly at multiple sites near the *c-myc* gene in HeLa cells. *Nucleic Acids Research* **24**, 1887–1894 (1996).
18. Sutherland, C., Cui, Y., Mao, H. & Hurley, L. H. A Mechanosensor Mechanism Controls the G-Quadruplex/i-Motif Molecular Switch in the *MYC* Promoter NHE III₁. *J. Am. Chem. Soc.* **138**, 14138–14151 (2016).
19. Sun, D. & Hurley, L. H. The Importance of Negative Superhelicity in Inducing the Formation of G-Quadruplex and i-Motif Structures in the *c-Myc* Promoter: Implications for Drug Targeting and Control of Gene Expression. *J. Med. Chem.* **52**, 2863–2874 (2009).
20. Langston, L. D. & O’Donnell, M. E. An explanation for origin unwinding in eukaryotes. *eLife* **8**, e46515 (2019).
21. Esain-Garcia, I. *et al.* G-quadruplex DNA structure is a positive regulator of *MYC* transcription. *Proc. Natl. Acad. Sci. U.S.A.* **121**, e2320240121 (2024).

22. Sengupta, P., Chattopadhyay, S. & Chatterjee, S. G-Quadruplex surveillance in BCL-2 gene: a promising therapeutic intervention in cancer treatment. *Drug Discovery Today* **22**, 1165–1186 (2017).
23. Chen, L. *et al.* Structural basis for nucleolin recognition of MYC promoter G-quadruplex. *Science* **388**, eadr1752 (2025).
24. Sengupta, P. & Chatterjee, S. Inosine 5'-diphosphate, a molecular decoy rescues Nucleoside diphosphate kinase from c-MYC G-Quadruplex unfolding. *Biochimica et Biophysica Acta (BBA) - General Subjects* **1864**, 129649 (2020).
25. Hänsel-Hertsch, R., Spiegel, J., Marsico, G., Tannahill, D. & Balasubramanian, S. Genome-wide mapping of endogenous G-quadruplex DNA structures by chromatin immunoprecipitation and high-throughput sequencing. *Nat Protoc* **13**, 551–564 (2018).
26. Zanin, I. *et al.* Genome-wide mapping of i-motifs reveals their association with transcription regulation in live human cells. *Nucleic Acids Research* **51**, 8309–8321 (2023).
27. Boissieras, J. *et al.* iMab antibody binds single-stranded cytosine-rich sequences and unfolds DNA i-motifs. *Nucleic Acids Research* **52**, 8052–8062 (2024).
28. Shi, R., Wang, Y., Yang, Q. & Li, F. Selective Recognition and Ultra-Sensitive Detection of i-Motif DNA Using a Novel Fluorescent Probe with PCA-Enhanced Quantification. *Anal. Chem.* **97**, 14693–14699 (2025).
29. Školáková, P. *et al.* Systematic investigation of sequence requirements for DNA i-motif formation. *Nucleic Acids Research* **47**, 2177–2189 (2019).
30. Guneri, D. *et al.* Structural insights into i-motif DNA structures in sequences from the insulin-linked polymorphic region. *Nat Commun* **15**, 7119 (2024).
31. Dai, J., Hatzakis, E., Hurley, L. H. & Yang, D. I-Motif Structures Formed in the Human c-MYC Promoter Are Highly Dynamic—Insights into Sequence Redundancy and I-Motif Stability. *PLoS ONE* **5**, e11647 (2010).
32. Li, T. & Famulok, M. I-Motif-Programmed Functionalization of DNA Nanocircles. *J. Am. Chem. Soc.* **135**, 1593–1599 (2013).
33. Bag, S. *et al.* Exploring i-Motif DNA binding with benzothiazolino Coumarins: Synthesis, Screening, and spectroscopic insights. *Bioorganic Chemistry* **156**, 108227 (2025).
34. Abdelhamid, M. A. S. & Waller, Z. A. E. Tricky Topology: Persistence of Folded Human Telomeric i-Motif DNA at Ambient Temperature and Neutral pH. *Front. Chem.* **8**, 40 (2020).
35. Rajendran, A., Nakano, S. & Sugimoto, N. Molecular crowding of the cosolutes induces an intramolecular i-motif structure of triplet repeat DNA oligomers at neutral pH. *Chem. Commun.* **46**, 1299 (2010).

Reviewer #1 (Remarks to the Author):

The authors have addressed all major comments.

Minor comment: the affinity of PCBP1 for unstructured C-rich DNA. While this affinity is consistently lower compared to iM structures, and this difference has been confirmed through various methods, it remains notably high, lowering the impact of the process. This observation warrants further consideration,

Response: Thank you for this comment. To highlight this difference, we have now included the following section in the discussion. ‘We also demonstrate that PCBP1 retains appreciable affinity for unstructured C-rich DNA, indicating a degree of binding plasticity that may be relevant under physiological conditions where i-motif stability fluctuates. Nonetheless, quantitative and competitive analyses demonstrate a consistent energetic and competitive advantage for folded i-motifs, suggesting that i-motif stability and C-density together modulate PCBP1 binding.’

Reviewer #2 (Remarks to the Author):

The authors have addressed all my previous comments thoroughly and have substantially strengthened the manuscript. The resubmission is significantly improved in clarity, methodological rigor, and contextual grounding within the existing i-motif literature. The new experiments, additional controls, and revised analyses convincingly support the conclusions. I find the paper suitable for publication in Nature Communications.

Reviewer #3 (Remarks to the Author):

Reviewer #4 (Remarks to the Author):

The additional experiments carried out by the authors fully address my concerns. The work undertaken to respond to the reviewers’ comments is of very high quality, demonstrating a sound scientific approach and strong supporting arguments. Consequently, I support the publication of this article in Nature Communications.

Reviewer #5 (Remarks to the Author):

I co-reviewed this manuscript with one of the reviewers who provided the listed reports. This is part

of the Nature Communications initiative to facilitate training in peer review and to provide appropriate recognition for Early Career Researchers who co-review manuscripts.

Response: We thank all the reviewers for their time and valuable feedback.